# How to prepare your task head for finetuning

**Yi Ren**
University of British Columbia
renyi.joshua@gmail.com

**Shangmin Guo**
University of Edinburgh
s.guo@ed.ac.uk

**Wonho Bae**
University of British Columbia
whbae@cs.ubc.ca

**Danica J. Sutherland**
University of British Columbia & Amii
dsuth@cs.ubc.ca

## Abstract

In deep learning, transferring information from a pretrained network to a downstream task by finetuning has many benefits. The choice of task head plays an important role in fine-tuning, as the pretrained and downstream tasks are usually different. Although there exist many different designs for finetuning, a full understanding of when and why these algorithms work has been elusive. We analyze how the choice of task head controls feature adaptation and hence influences the downstream performance. By decomposing the learning dynamics of adaptation, we find that the key aspect is the training accuracy and loss at the beginning of finetuning, which determines the "energy" available for the feature's adaptation. We identify a significant trend in the effect of changes in this initial energy on the resulting features after finetuning. Specifically, as the energy increases, the Euclidean and cosine distances between the resulting and original features increase, while their dot products (and the resulting features' norm) first increase then decrease. Inspired by this, we give several practical principles that lead to better downstream performance. We analytically prove this trend in an overparamterized linear setting, and verify its applicability to different experimental settings.

## 1 Introduction

In the era of deep learning, pretraining a model on a large dataset and adapting it to downstream tasks is a popular workflow. With the help of large amount of data and huge computing resources, the pretrained model can usually provide beneficial features for the downstream tasks. Such a framework is proven to be efficient and effective in many domains and tasks, e.g. natural language processing (Kenton & Toutanova, 2019), computer vision (Chen et al., 2020b), graph based learning (Liu et al., 2022), and so on. Although different variants of pretraining and finetuning (FT) methods are widely applied – including direct finetuning, finetuning after linear probing (Kumar et al., 2022), side-tuning (Zhang et al., 2020a), using different learning rates for different layers (Zhang et al., 2021), and more – a detailed understanding of how features are adapted during finetuning under different settings remains elusive.

Our work builds significantly off the analysis of Kumar et al. (2022), who study the interactions between the "task head" (the final layer of the network, usually randomly initialized) and the "backbone" (usually copied from the pretrained model). Kumar et al. claim that the standard finetuning method, randomly initializing a task head then updating all parameters of the whole network, can distort the pretrained features and hence can deteriorate the generalization ability if (as they assume) the previous backbone features were optimal for the downstream task. By analyzing an overparameterized linear model, they prove that linear probing (i.e., only updating the parameters of the task head) first, followed by finetuning the whole network, leads to better performance in their setting.

In this work, we consider less stringent assumptions than they made, and study more practical settings from a different perspective. First, we consider scenarios where the pretrained features are not optimal for the downstream tasks, thus feature adaptation is indeed beneficial. Unlike the two extreme cases studied by Kumar et al. (2022), i.e. finetuning with fully random initialization and

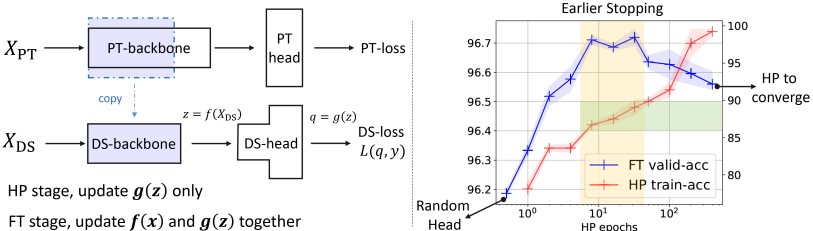

Figure 1: Left: a general example of pretraining (PT), head probing (HP) and finetuning (FT) procedure (DS is short for downstream). Right: an example showing that neither probing the head to converge nor no probing is the optimum (pretrained on ImageNet-1K and finetuned on STL10).

fully-pretrained parameters, we consider intermediate cases where features are mildly adapted by stopping earlier (before convergence) in the linear probing procedure. To better understand the feature's behavior, we decompose the learning dynamics of the feature vector during finetuning based on "energy" and "direction" of the learning. We discover a non-trivial trend in how this "energy" affects the way that features change from their initialization, which can inspires us to design an appropriate finetuning procedure. Under this framework, we demonstrate that the "unchanged feature" assumption of Kumar et al. (2022) is hard to achieve.

Second, our task heads are not necessarily linear. Inspired by the illustrations of Olah et al. (2020), it is reasonable to only preserve the lower layers of the pretrained model and reinitialize the top layers, assuming that the low-level features are common across task. That is, the probed task head is non-linear, and we refer to this more general process as "head probing" (HP) rather than linear probing. Our analysis can also help to explain feature behavior in this setting.

Finally, following our analysis, we provide a user guide to conclude when and why specific methods should be considered. Specifically, we have one basic method: stop head probing earlier, **before** convergence; and three advanced tricks: 1.) use label smoothing during head probing; 2.) use more complex task head design; 3.) merge and reinitialize some later layers of the backbone and attach them to the task head. In summary, in this work:

- we formalize and explain feature adaptation by decomposing the learning dynamics;
- we find a non-trivial trend in feature adaptation and verify it in many cases;
- and we show how controlling feature adaptation can improve downstream performance.

## 2  MOTIVATION

Pretrain-then-finetune is a popular workflow for many tasks in deep learning. One common practice is to 1) randomly initialize a task head, 2) attach it to a pretrained backbone, then 3) finetune the whole network together (Li et al., 2020). However, the untrained task head may distort the pretrained features during finetuning. To solve this problem, Kumar et al. (2022) propose to train the head to fully converge before the finetuning. However, suppose we train the head long enough and its training accuracy (HP-train-acc) converges to 100%, then the features won't change during the finetuning stage. To sum up from the above, we can see that neither probing the head to converge nor no probing is optimal, since the pretraining and downstream tasks (or datasets) are usually distinct. To verify this argument, we HP various number of epochs before finetuning, and record the corresponding validation accuracy after finetuning (FT-valid-acc for short), and the results are shown in Figure 1. It is surprising to see that stopping the head training earlier (before the convergence of HP-train-acc) brings more improvement. As the only variable among these experiments is the parameters of the head before finetuning, the following two questions emerge:

*How does the task head influence the pretrained features during finetuning?*
*How does the feature adaptation influence the generalization performance after finetuning?*

## 3  BACKGROUND

We first clarify the scope of our analysis. We don't consider the details of the pretraining procedure, instead just assuming that there are some well-trained checkpoints for a particular dataset or task.

Meanwhile, our formulation is not restricted to classification tasks; our use of the term "label" or "target" can be any form of supervisory signals.

## 3.1 PROBLEM SETTING AND TWO STAGE TRAINING

When training a model for a downstream task, our goal is to find a predictor $f \circ g : \mathbb{R}^d \to \mathbb{R}^k$ that maps a high-dimensional input signal $\mathbf{x} \in \mathbb{R}^d$ to a task-related prediction $\mathbf{q} \in \mathbb{R}^k$. As depicted by the left-bottom panel of Figure 1, we split the predictor into two parts: the backbone $f(\mathbf{x}; \mathbf{B})$ : $\mathbb{R}^d \to \mathbb{R}^h$ which maps the input signal to intermediate representations $\mathbf{z} \in \mathbb{R}^h$, and the task head $g(\mathbf{z}; \mathbf{v}) : \mathbb{R}^h \to \mathbb{R}^k$ which gives the prediction vector $\mathbf{q} \in \mathbb{R}^k$ (e.g. logits in classification tasks using cross-entropy loss). Usually, the backbone $f$ is parameterized by $\mathbf{B}$ and initialized by copying from a pretrained model. The task head $g$ parameterized by $\mathbf{v}$, on the other hand, is usually randomly initialized and might be a complex non-linear function. The training has two distinct stages: 1) head probing (HP) where we fix $f$ and only update the parameters $\mathbf{v}$ of $g$ for $\tau$ epochs; 2) finetuning (FT) where the parameters $\{\mathbf{v}, \mathbf{B}\}$ of $f \circ g$ are updated together until convergence. In this work, we analyze how the FT stage is influenced by the architecture and $\mathbf{v}$ at the beginning of finetuning.

Following the general formulation above, we show a simple overparameterized linear regression (or equivalently, a binary classification) below as a case study to illustrate more insights. Suppose the $N$ input signals are stacked, $X = [\mathbf{x}^{(1)}, ..., \mathbf{x}^{(N)}]^\top \in \mathbb{R}^{N \times d}$, the loss function can be written as

$$\mathcal{L}_{\mathbf{B}, \mathbf{v}} = \frac{1}{2} \|X\mathbf{B}^\top \mathbf{v} - Y\|_2^2, \tag{1}$$

where $\mathbf{B} \in \mathbb{R}^{h \times d}$, $\mathbf{v} \in \mathbb{R}^h$, and $Y \in \mathbb{R}^N$. That is, $k = 1$, $\mathbf{z} = \mathbf{B}\mathbf{x}$, $q = \mathbf{z}^\top \mathbf{v}$, and $\mathcal{L}(q, y) = \frac{1}{2}(q - y)^2$.

## 3.2 WHAT TO EXPECT DURING ADAPTATION

Compared with training a randomly initialized model, adaptation on downstream tasks needs more care. One reason is that the pretrained parameters from $f(\mathbf{x}; \mathbf{B})$ inherits all the information from the pretraining task, even any bias and noise. Furthermore, as mentioned by Kumar et al. (2022) and Du et al. (2018), $f(\mathbf{x}; \mathbf{B})$ is tied to $g(\mathbf{z}; \mathbf{v})$ at each time step during FT, thus the bias and noise in $f(\mathbf{x}; \mathbf{B})$ also influence the learning of $g(\mathbf{z}; \mathbf{v})$. Hence, before conducting downstream adaptation, we might consider: to what extent do we *want* to change the feature extractor $f(\mathbf{x}; \mathbf{B})$? Here, we list three possible cases of how much we should update $f(\mathbf{x}; \mathbf{B})$:

- Strong: the pretrained features are far from the optimal ones for downstream task, so we need substantial feature adaptation.
- Mild: $f(\mathbf{x}; \mathbf{B})$ is reasonably good, but adaptation to the downstream domain is helpful.
- Tiny: the pretrained $f(\mathbf{x}; \mathbf{B})$ is near optimal and only need to be slightly adapted.

In the rest of this paper, we first analyze how $f(\mathbf{x}; \mathbf{B})$ and $g(\mathbf{z}; \mathbf{v})$ interact with each other, under both the general case and the simplified case. Based on our observations, we propose several practical principles in Section 5.1 for better downstream performance.

## 4 INTERACTION BETWEEN THE BACKBONE AND HEAD

Rather than linking the choice of $\mathbf{v}_0$ (the task head parameters at the beginning of FT) to the loss function, we sketch how $f(\mathbf{x}; \mathbf{B})$ and $g(\mathbf{z}; \mathbf{v})$ interact during FT, which depicts how $\mathbf{z}$ changes accordingly. Although our analysis cannot provide any theoretical guarantee, knowing *how $\mathbf{z}$ changes under different $\mathbf{v}_0$* will lead us to better HP-FT design in practice.

## 4.1 AVERAGE INITIAL ENERGY

We start from analyzing the behavior of $\mathbf{z}_t^{(j)} = f(\mathbf{x}^{(j)}; \mathbf{B}_t)$, i.e. the feature extractor at time $t$, when the network parameters are updated with samples $\mathbf{x}^{(1)}, ..., \mathbf{x}^{(N)}$ using gradient descent. When using

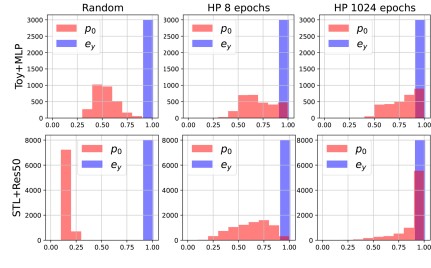

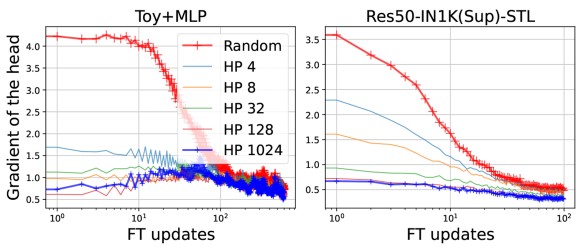

(a) Influence of $\tau$ on energy.

(b) Influence of $\tau$ on direction.

Figure 2: Left: histogram of $\mathbf{p}_0$ and $\mathbf{e}_y$ under different $\tau$. Right: approximated change of the 'direction' term in Equation (2) under different $\tau$. The titles represent the settings. HP $\tau$ means HP for $\tau$ epochs. Toy+MLP means pretrain a 4-layer MLP on full MNIST then transfer to a distorted subset of MNIST. Res50-IN1K(Sup)-STL means pretrain a ResNet50 on ImageNet-1k using supervised classification, then transfer to a downstream task on STL10. See Appendix B for more details.

cross-entropy loss, we can have the following result: [1]

$$
\mathbf{z}_{t+1}^{(j)} - \mathbf{z}_t^{(j)} = \frac{\gamma}{N} \sum_{n=1}^{N} \left( \underbrace{\kappa_t^{(j,n)}}_{\text{slow-change}} \cdot \underbrace{\left( \nabla_{\mathbf{z}} \mathbf{q}_t^{(n)} \right)^{\top}}_{\text{direction}} \cdot \underbrace{\left( \mathbf{e}_{y_n} - \mathbf{p}_t^{(n)} \right)}_{\text{energy}} \right) + \mathcal{O}(\gamma^2), \tag{2}
$$

where $\gamma$ is the learning rate, $\kappa_t^{(j,n)} = \left( \nabla_{\mathbf{B}} \mathbf{z}_t^{(j)} \right) \left( \nabla_{\mathbf{B}} \mathbf{z}_t^{(n)} \right)^{\top} \in \mathbb{R}^{h \times h}$ is the empirical neural tangent kernel (NTK) of the backbone between $\mathbf{x}^{(j)}$ and $\mathbf{x}^{(n)}$ at time $t$,[2] $\nabla_{\mathbf{z}} \mathbf{q}_t^{(n)} \in \mathbb{R}^{k \times h}$ is the gradient of the task head prediction w.r.t the representation vector $\mathbf{z}$ at time $t$,[3] $\mathbf{p}_t^{(n)} = \text{Softmax}(\mathbf{q}_t^{(n)})$ is the predicted probability vector for input $\mathbf{x}^{(n)}$, and $\mathbf{e}_{y_n}$ is the one-hot vector of the label $y_n$.

In this decomposition, the first term is often understood to change only slowly during FT, i.e. the lazy-parameters setting used by Chizat et al. (2019); Yang & Hu (2020). The second term determines the direction, and the last term provides "energy" for the update of $\mathbf{z}$. Formally, we can define the Average Initial Energy (AIE) and use it to bound the norm of $\mathbf{z}_T^{(j)} - \mathbf{z}_0^{(j)}$:

**Proposition 1.** $\mathbb{E}_{\mathbf{x}^{(j)}} \| \mathbf{z}_T^{(j)} - \mathbf{z}_0^{(j)} \|_2 \le c \cdot E_{aie}$, where $E_{aie} \triangleq \mathbb{E}_{\mathbf{x}^{(n)}} \| \mathbf{e}_{y_n} - \mathbf{p}_0^{(n)} \|_2$ is the Average Initial Energy (AIE). Here $c$ is a constant, $T$ is the FT epochs, and $\mathbf{p}_0^{(n)}$ is the model's prediction of sample $\mathbf{x}^{(n)}$ at the beginning of FT (or at the end of HP).

*Proof.* See Appendix A. □

Although this bound is loose and requires approximations based on assumptions, the proposition supports our intuition: if we prob the head for too long (i.e. a large $\tau$), the HP-train-acc can be very high, hence on average $\mathbf{p}_0$ is close to $\mathbf{e}_y$ and the features adapt less, and vice versa. Our intuition is further verified in the right panel of Figure 2 where we plot the predicted probability of the correct class, i.e. $[\mathbf{p}_0^{(n)}]_{y_n}$, for each sample. Furthermore, we find that it is unlikely to obtain zero energy (not as assumed by Kumar et al. (2022)) in many practical applications: even with high accuracy there will always be some gap between $\mathbf{p}_0$ and $\mathbf{e}_y$ by definition, and moreover the training accuracy after HP is sometimes far from perfect (see Section 5).

### 4.2 NON-TRIVIAL TREND OF FEATURE ADAPTATION

Following Proposition 1, we can link the adaptation of features to $\tau$ via AIE. In the proof of Proposition 1, we disentangle the dependence of the direction and energy terms using Cauchy-Schwarz

---

[1] The case for MSE loss, as well as the derivation of this equation, can be found in Appendix A.

[2] In a linear model $\mathbf{z} = \mathbf{B}\mathbf{x}$, $\kappa_t^{(j,n)} = (\mathbf{x}^{(j)})^{\top}(\mathbf{x}^{(n)}) \cdot I_{h \times h}$ is invariant during FT (i.e., independent of $t$).

[3] Note that $\nabla_{\mathbf{z}} \mathbf{q}_t^{(n)}$ depends on the parameters of the current task head $\mathbf{v}_t$ in the generally, whereas $\nabla_{\mathbf{z}} \mathbf{q}_t^{(n)} = \mathbf{v}^{\top}$ is independent of $n$ if the model is linear.

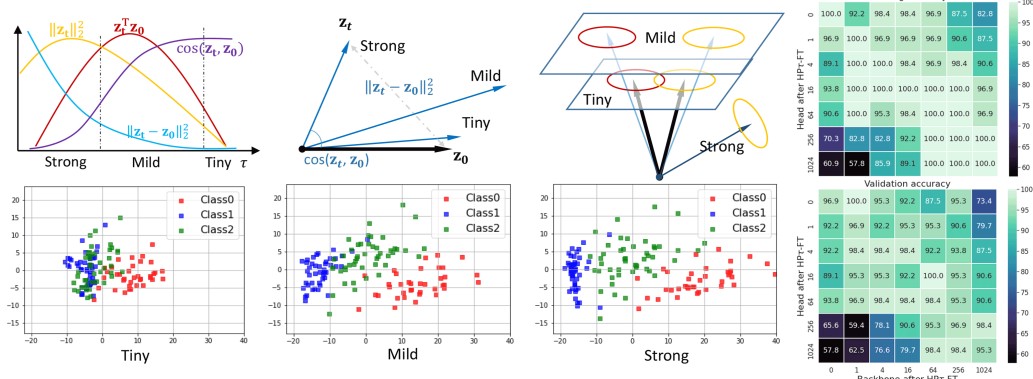

Figure 3: How $\mathbf{z}_t$ changes from $\mathbf{z}_0$. The ellipses in the third panel represent the scattered features. The scatter plots are the PCA projection of the resulting $\mathbf{z}_t$ after finetuning. For tiny, mild, and strong adaption, we use $\tau = 1024, 64, 0$, respectively. The heat-maps of the last column are the "head-exchange" experiments (all under the Toy+MLP settings).

inequality, i.e., $(\nabla_{\mathbf{z}}\mathbf{q})^{\top}(\mathbf{e}_y - \mathbf{p}_0) \le \|\nabla_{\mathbf{z}}\mathbf{q}\|_2 \cdot \|\mathbf{e}_y - \mathbf{p}_0\|_2$. However, recall that the direction term Equation (2) in also plays an important role, especially at the beginning of finetuning.

To verify our hypothesis about the direction term, we depict the change of this term during finetuning in the right panel of Figure 2. Suppose a linear task-head (non-linear heads have similar behavior in the NTK regimes), this quantity can be approximated by the norm of the gradients to $\mathbf{v}_t$, since $\|\nabla_{\mathbf{z}}\mathbf{q}_{t+1} - \nabla_{\mathbf{z}}\mathbf{q}_t\|_F^2 = \|\mathbf{v}_{t+1}^{\top} - \mathbf{v}_t^{\top}\|_2^2 = \gamma^2 \|\nabla_{\mathbf{v}}\mathcal{L}\|_2^2$. As we find $\|\mathbf{v}_t\|_2$ changes little during the finetuning stage, the large $\|\nabla_{\mathbf{v}}\mathcal{L}\|_2$ is more likely from a big direction change. As illustrated by Figure 2, when $\tau = 0$, $\nabla_{\mathbf{z}}\mathbf{q}_t$ changes a lot at the beginning of FT, which can make $\mathbf{z}^{(n)}$ change in inconsistent directions. When $\tau = 1024$, the direction term changes only a little through finetuning.

This finding inspires us to look deeper at what is the difference between a strong adaptation (e.g., $\tau = 0$) and a mild adaptation (e.g., $\tau = 4$). To get an overall picture of $\mathbf{z}$'s change, only $\|\mathbf{z}_T - \mathbf{z}_0\|_2$ is not enough. Hence we analyze the following four quantities related to the similarity between the features before finetuning $\mathbf{z}_0$ and afterwards $\mathbf{z}_T$: $\|\mathbf{z}_T - \mathbf{z}_0\|_2$, $\|\mathbf{z}_T\|_2$, $\mathbf{z}_T^{\top}\mathbf{z}_0$, and $\cos(\mathbf{z}_T, \mathbf{z}_0)$. With an overparameterized model, we can analytically calculate the expressions of them and make the following conclusion:

**Proposition 2** (Informal). *In an overparameterized two-layer linear model, when $\tau$ increases (the AIE decreases), $\|\mathbf{z}_T - \mathbf{z}_0\|_2^2$ monotonically decreases while $\|\mathbf{z}_T\|_2^2$ and $\mathbf{z}_T^{\top}\mathbf{z}_0$ exhibit a **quadratic** trend. The trend of $\cos(\mathbf{z}_T, \mathbf{z}_0)$ is hard to predict, but there is a phase that this value increases fast.*

Such a trend is illustrated in Figure 3, supported by various experiments in Appendix C, and strictly proved in Appendix D. With this information, we can infer the behavior of $\mathbf{z}_T$ and sketch it in the second and the third panels. In the "tiny" case, when $\mathbf{v}_0^{\tau}$ is fully converged ($\tau \to \infty$), the features are almost stable during finetuning; this only works well when the pretrained features are perfect for the downstream task. In the "mild" case, when $\tau$ is reasonably large, the resulting $\mathbf{z}_T$ will be stretched ($\|\mathbf{z}_T\|_2$ increases) in a similar direction (cosine changes little). This kind of $\mathbf{z}$ can make the original overlapped features become more separable without changing the manifold of the features too much, which is desirable in most transfer learning scenarios. For the "strong" case, where we only HP for a few updates or simply use a random head, $\mathbf{z}_T$ will change in an unpredictable way, especially for the early updates. Although the fine-tuned model may generalize well, $\mathbf{z}_T$ might be quite different from $\mathbf{z}_0$. Thus, if we believe the pretrained features are too specific to the pretraining dataset, the "strong" case is a reasonable choice.

Note that even though the scatter plots of the mild and strong cases look similar, the corresponding feature manifold might be quite different. To verify this, we first run HP$\tau$-FT (i.e., load the pretrained model, prob the head for $\tau$ epochs, then finetune to converge) for 7 different $\tau$. Then we save the converged backbone and task-head separately for each $\tau$, and pair-wisely exchange the head and backbone to build 49 new models (without further tuning). The training and validation accuracy of the 49 models are reported in Figure 3. The results match well with our analysis: 1) the

off-diagonal values are lower, which means the representations learned by experiments with different $\tau$ are different; 2) for the strong adapted backbones (the first two columns), the heads from other cases are not compatible, which means the features' manifold are significantly changed; 3) for the mild and tiny cases (latter columns), the aforementioned incompatibility almost disappears, which means the features' manifold of these cases are quite similar.

## 4.3 BACKBONE DEPTH AND HEAD CAPACITY

Beyond how long we train the task head, the structure and capacity of the task head also influence the training prediction after this stage, thus the adaptation of $\mathbf{z}_t$. We briefly discuss the trend here and verify them in Section 5. When using a low-capacity head (e.g. a linear head parameterized by a $h \times k$ matrix), it might be hard to preserve the pretrained $\mathbf{z}_0$ even a very large $\tau$ is chosen, as the head cannot achieve a high enough training accuracy to decrease AIE. On the other hand, if the capacity of the task head is much bigger than the backbone, the information from the pretrained network might be easily distorted. For example, consider using only the first block of a pretrained ResNet18 as the backbone and concatenating the other 3 blocks with a 10-layer wide MLP. If this huge task head is randomly initialized, the information contained in the backbone could be completely rewritten, as the random changing phase of $\mathbf{z}_t$ can be very long (remember that in Equation (2), $\mathbf{z}_t$ will change in random directions before $\nabla_{\mathbf{v}}\mathbf{q}_t$ becomes less noisy).

## 5 EXAMPLES ON REAL TASKS

In this section, we first provide a "user guide" of all the aforementioned methods. Note that given the pretrained and downstream tasks, determining the optimum adaptation energy is rather heuristic: the user guide only provides some high-level suggestions for the practitioners. We then provide abundant real examples of how to apply these principles. Generally, we can first try sweeping the optimal $\tau$ using linear head and consider other advanced tricks if necessary.

## 5.1 USER GUIDE

The goal of this paper is to provide a toolbox for preparing the task-head before finetuning. Here, we suggest a "*phenomenon → hypothesis → solution*" workflow and use the validation performance for verification. Recall how we mitigate the overfitting problem using dropout: *validation accuracy decrease after convergence → model overfitted → add dropout*. Similarly, we can have: *HP train-acc converge to 100% → no enough energy → use smaller $\tau$*.

However, in practice, it is unknown that how much energy is beneficial, as the neural network might not encode the input samples as we humans expected. Hence, we suggest starting from the basic setting, i.e. using a linear head and sweeping $\tau$ to get a high-level understanding of how the energy influence the downstream generalization. Usually, selecting the optimal $\tau^*$ using validation accuracy can ensure a reasonably good test performance, as verified in the next subsection.

The advanced tricks are only applicable to specific scenarios and need more consideration. For example, if we really want tiny energy but using the linear head only achieves less than 50% training accuracy after head probing, we can consider an MLP head to increase the converged training accuracy (hence reduce the energy). If we want a mild adaptation, but the training accuracy during HP goes to 100% too fast, using label smoothing during HP can be considered. If we believe the downstream task only needs low-level features of the backbone, partial-backbone is a good choice. However, these advanced tricks also have side effects that are deleterious to the downstream performance. We will analyze their advantage and limitations in Section 5.3 with concrete examples.

## 5.2 BASIC METHOD: EARLIER STOPPING HP

Following our theory, when the learning rate for HP is fixed, the value of $\tau$ is positively correlated with the training accuracy, thus negatively correlated with energy. Hence, given a pretrained backbone and a downstream task, we can always start from sweeping $\tau$ (see Figure 4).

As stated in the motivation part, during finetuning, we usually expect some feature adaptation while keeping some pretrained information. Hence, neither $\tau = 0$ nor $\tau = \infty$ is optimal. We find such a

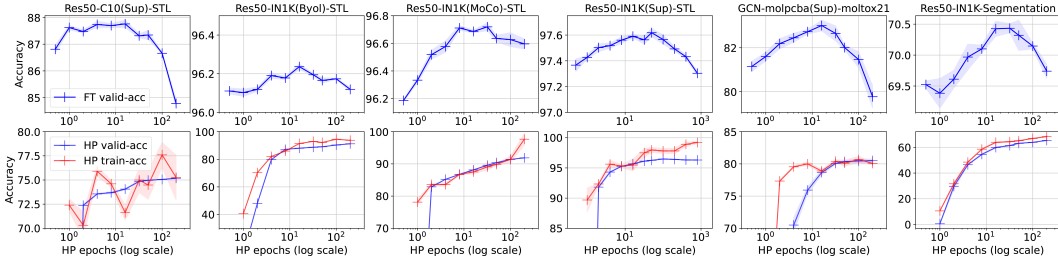

Figure 4: Sweeping $\tau$ from 0 to 200 (in $2^n$ fashion). The valid-accuracy of FT-only setting is the left most point in each panel. The first 4 columns are on image classification task, the fifth one is on graph task, and the last one is on image segmentation task.

| downstream data: | Flowers | | | STL10 | | | CIFAR100 | | | Tox21 |
|---|---|---|---|---|---|---|---|---|---|---|
| pretraining task-data: | MoCo-IN | Sup-IN | Sup-C10 | MoCo-IN | Sup-IN | Sup-C10 | MoCo-IN | Sup-IN | Sup-C10 | Sup-pcba |
| HP-train-acc | 59.711 | 85.826 | 7.031 | 91.889 | 96.291 | 75.181 | 56.397 | 62.709 | 11.735 | 79.121 |
| HP0-FT | 76.953 | 91.295 | 60.714 | 96.136 | 97.452 | 86.811 | 80.432 | 84.736 | 65.608 | 81.274 |
| HP200-FT | 84.882 | 91.653 | 43.973 | 96.363 | 97.374 | 84.778 | 83.744 | 84.347 | 63.992 | 79.666 |
| HP$\tau^*$-FT | **86.831** | **92.299** | **63.711** | **96.753** | **97.697** | **87.739** | **83.814** | **84.971** | **65.966** | **83.263** |
| lsHP200-FT | 86.946 | 92.746 | 42.299 | 96.639 | 97.959 | 85.912 | 83.412 | 85.357 | 65.244 | 83.853 |

Table 1: Downstream test accuracy across different settings. $\tau^*$ is selected based on validation accuracy in sweeping. The HP$\tau^*$ always bring some improvement. The lsHP can further improve when the HP-only method achieves a high accuracy (no enough energy for adaptation), but fails when the pretraining features are not suitable (in Sup-C10 case). See Section 5.3 for more discussions.

trend is consistent across various settings: from image input to graph input, from classification task to segmentation task, and from supervisory pre-training to unsupervised pre-training.

Other than this general trend, the nuance of experiments under different settings also supports our analysis well. Specifically, in the image classification experiments shown in Figure 4, adapting a ResNet50 to STL10 behaves differently on different pretraining task. In the first column, we see a large $\tau$ (small energy) hurts the downstream generalization performance, because the features pretrained on CIFAR10 might be far from optimal for the downstream task. In contrast, the features pretrained on ImageNet-1K (IN1K for short) all provide good results. Among those IN1K pretrained models, the model from a supervised classification task leads the best overall downstream performance, but large $\tau$ is still harmful. In other words, the features pretrained this way might be somewhat too specific to the pretraining dataset, and hence mild adaptation is beneficial. Regarding the unsupervised pretraining cases, BYOL (Grill et al., 2020) is less sensitive to the choice of $\tau$, while MoCo (He et al., 2020) behaves more similarly to the supervised case.

Another interesting finding is that the HP training accuracy at the optimal $\tau^*$ is usually smaller than the converged value: we cannot select $\tau$ based on the standard early stopping criterion on HP train accuracy. As the task head under $\tau^*$ usually has not converged on the pretraining dataset, we call this method "earlier stopping HP" (HP$\tau^*$-FT for short). As shown in Table 1, HP$\tau^*$-FT can always bring improvements.

## 5.3 ADVANCED TRICKS: SUPERIORITY AND LIMITATION

**MLP Head:** Instead of using a linear head, authors of Chen et al. (2020b) claim that using an MLP head can sometimes bring improvement. Following the analysis of this paper, we can consider this trick when we want small energy while the linear head cannot reach a high HP training accuracy even after a long HP. For example, in the first two columns in Table 2, the HP-train-acc in the linear head cases plateaued after reaching 92% or 78%, while a 2-layer MLP can reach 99% and 95%. As the energy decreases, features adapt less during finetuning (see the decreased distance metrics) and the final models generalize better.

However, we should be careful when applying this trick if we want to use a small $\tau$ (i.e., large energy). Recall our analysis that the inconsistent direction term (i.e., $\nabla_{\mathbf{z}}\mathbf{q}_t$ defined in Equation (2)) makes the feature adaptation more unpredictable at the beginning of finetuning. Increasing the head capacity in this case would make the head converge slower and hence prolong such a chaos phase.

| | | Sim-Real | Sup-Sketch | | Sim-STL | Sup-STL | | Sim-STL | Sup-STL |
|---|---|---|---|---|---|---|---|---|---|
| HP-train-acc | | 92.676 | 78.213 | Baseline | 96.875 | 100 | Small energy $\eta_{FT}=0.9$ $\eta_{HP}=0.9$ | 97.607 | 100 |
| $1-\cos(\mathbf{z}_T,\mathbf{z}_0)$ | Baseline | 0.1269 | 0.1422 | Small energy | 0.0044 | 0.017 | | 0.0049 | 0.0256 |
| $\|\mathbf{z}_T-\mathbf{z}_0\|_2^2$ | Linear-head | 5.247 | 4.752 | $\eta_{FT}=1$ | 1.792 | 6.716 | | 1.731 | 7.391 |
| FT-val-acc | | 78.075 | 61.545 | $\eta_{HP}=1$ | 93.914 | 97.581 | | 94.015 | 97.694 |
| HP-train-acc | | 99.121 | 94.883 | More energy | 96.875 | 100 | Opposite Energy $\eta_{FT}=0.9$ $\eta_{HP}=1$ | 97.982 | 100 |
| $1-\cos(\mathbf{z}_T,\mathbf{z}_0)$ | 2MLP-head | 0.1098 | 0.1104 | $\eta_{FT}=1$ | 0.014 | 0.0549 | | 0.0109 | 0.0849 |
| $\|\mathbf{z}_T-\mathbf{z}_0\|_2^2$ | | 4.942 | 4.206 | $\eta_{HP}=0.9$ | 1.841 | 9.278 | | 3.041 | 13.881 |
| FT-val-acc | | 78.453 | 63.773 | | 94.304 | 97.92 | | 93.208 | 97.039 |

Table 2: How ls-HP and larger head influence the feature adaptation. The models are pretrained using SimCLR (Sim) or supervised (Sup) classification on IN1K. Number in blue (red) represent an decrease (increase) compared with its counterpart in the baseline. See more results in Appendix C.

**Label Smoothing HP:** Recall that the energy is upper bounded by $\|\mathbf{e}_y - \mathbf{p}_0\|_2^2$, where $\mathbf{p}_0$ is the model's prediction after HP for $\tau$ epochs. $\mathbf{p}_0$ converges to the labels used in HP when $\tau \to \infty$. Hence, instead of changing $\tau$, we can also manipulate the labels in HP to achieve a similar goal. One simple yet effective way is label smoothing (Müller et al., 2019, e.g.). By setting the labels during HP as $\eta_{HP}\mathbf{e}_y + (1 - \eta_{HP}) * \mathbf{u}$, where $\mathbf{u}$ is a uniform K-class categorical distribution, the HP stage can always reserve at least $(1 - \eta_{HP}) * \|\mathbf{e}_y - \mathbf{u}\|_2$ energy for the following feature adaptation, even $\tau \to \infty$. Such a trick (lsHP for short) is quite helpful when the HP-train-acc converges to 90%+ very fast, yet we still want a mild adaptation, like the example shown in the second two columns in Table 2. With lsHP, we see that the features adapt more even the HP-train-accs are unchanged.

To verify that the aforementioned improvement comes from the reserved energy during lsHP, we further try using smoothed labels during finetuning (e.g., $\eta_{FT} = 0.9$). The results match our analysis: when $\eta_{HP} = \eta_{FT} = 0.9$, the reserved energy disappears, as the labels of the two phases are the same again. Hence, all the numbers under this condition are quite similar to the baseline case ($\eta_{HP} = \eta_{FT} = 1$). For the "opposite energy" case, we observe a larger adaptation but a worse generalization performance. That is because the reserved energy make the features adapt in opposite directions. These results remind us that if we decide to use smooth label in both finetuning and head probing (e.g., we assume most of the samples in the downstream dataset are ambiguous), we need to set $\eta_{HP} \le \eta_{FT}$ to ensure a correct direction.

In summary, the lsHP trick is suitable for scenarios where the pretrained features are pretty well and the standard HP converges to 90%+ very fast. When the HP-train-acc is too low, the assumption used in HP, i.e. $\mathbf{p}_0$ converges to the labels, no longer holds. Hence, lsHP does not always bring enhancement, an example is given in the last row in Table 1.

**Partial Backbone:**

This is a more intricate trick that requires a stronger and more heuristic hypothesis. For example, there is a common belief in the deep-learning community that the lower layers extract fundamental features (e.g. edges, curves, and texture in vision tasks) while the higher layers learn more semantic features (e.g. dog heads or wheels) (Baldock et al., 2021; Olah et al., 2020). Hence, if we believe the downstream task treats the low-level features as beneficial while the high-level features are harmful, reinitializing the higher layers (like the fortuitous forgetting mentioned by Zhou et al. (2022)) and incorporating them as part of the task head can be beneficial.

| | Real | Sketch | Quick |
|---|---|---|---|
| HP-train-acc | 92.676 | 78.213 | 64.307 |
| -L4.3 | 100 | 100 | 97.666 |
| -L4.2, -L4.1 | 100 | 100 | 100 |
| FT-val-acc | 78.075 | 61.545 | 59.703 |
| -L4.3 | **78.528** | **65.683** | 67.011 |
| -L4.2 | 78.427 | 65.336 | 67.087 |
| -L4.1 | 76.689 | 65.017 | **68.07** |

Table 3: ResNet50 pretrained on IN1K using SimCLR. -L4.x means reinitializing the blocks in the resnet (merge them to task head) until layer 4.x.

See the results in Table 3, where the Quick[4] dataset is likely to rely more on the low-level features learned during pretraining. So the optimal setting for the Quick column is removing information from the last three layers (i.e., L4.3, L4.2, and L4.1) in ResNet50, while the optimal setting for the other two cases is to reinitialize only the last layer (i.e., L4.3). Using this trick, the task-head capacity might increase significantly (see the HP-train-acc increase to 100% after reinitializing). Hence, the principles discussed in the MLP-head trick also hold here. However, as it is hard to figure out what are the beneficial features for downstream tasks, such a trick has the lowest priority in our toolbox.

---

[4]Refer to Figure 8 to get an intuition about what the samples in theses dataset look like.

## 6    RELATED WORK AND DISCUSSIONS

**HP, FT and HP-FT.** Head probing and fine tuning are two fundamental algorithms in transfer learning, which have also attracted much discussion and debate. Intuitively, by freezing the pretrained backbone, HP will re-combine the features without distorting them, and hence yields better performance than FT when features are perfect (Peters et al., 2019). However, when pretrain and downstream tasks are very different, adapting the features is important and FT outperforms HP (Chen et al., 2020b; Zhai et al., 2019; He et al., 2022). Combining the strengths of HP and FT, authors of Kumar et al. (2022) demonstrate that HP-FT (i.e., first HP, then FT) yields the best performance on both in-distribution and out-of-distribution cases. Although they provide some theoretical guarantees for the superiority of HP-FT, some of their assumptions are dubious. To shed more lights on the pros and cons of these two methods, this paper analyze in detail *how the features change* under different task-head settings. Specifically, by controlling the HP epochs $\tau$ or otherwise influencing the energy term, we can design different types of feature adaptations for different downstream tasks. The relationship between the choice of task head and resulting $\mathbf{z}$'s adaptation is explained using Equation (2), verified by various experiments, and proved in an overparameterized linear model. Although the paper only analyze these fundamental settings, we believe the analysis provided here can also be combined with (or explain the benefits of) other more complex finetuning mechanisms, like those of Guo et al. (2019); Zhang et al. (2020a); Aghajanyan et al. (2021); Howard & Ruder (2018). For example, a common practice in transfer learning is to use a small learning rate for the backbone and a large learning rate for task head. From our perspective, this method can weaken the influence of the "noisy direction" term and make the features adapt in a low energy condition.

**Backbone and head depth.** Besides the influence of the initial value of the head, this paper also discusses the influence of the relative capacity between backbone and head. Chen et al. (2020b) show that a simple MLP head sometimes also brings enhancement. When the backbone only copies the early layers of the pretrained network, as the downstream task might need different levels of features (Olah et al., 2020; Baldock et al., 2021; Zhang et al., 2020b), the later layers and the original head can be combined as the new complex task-head, like the example in Figure 1. Furthermore, in some encoder-decoder style models, like some the models for some language tasks (Peters et al., 2019; Zhu et al., 2019), the head (decoder) might have comparable capacity with the backbone (encoder). In another extreme case, if we use pretrained word2vec features (Mikolov et al., 2013) and plan to FT them in a downstream task, the task head is the whole network, which is much bigger than the embedding layer. We believe the discussion in this paper might also help inspire the design of HP-FT strategy for these practical scenarios.

## 7    CONCLUSION

This paper studies how the choice of task heads influence the pretrained features $\mathbf{z}$'s adaptation and hence influence the downstream performance. By decomposing the learning dynamics of $\mathbf{z}$, we find the keys are the energy and direction terms, which highly correlate with the accuracy at the beginning of the FT process (i.e., the accuracy after HP). Hence under a common HP-FT framework, we carefully analyze the relationship between the HP epochs ($\tau$) and the features' adaptation (discrepancy between $\mathbf{z}_T$ and $\mathbf{z}_0$). Different from most existing works, which mainly focus on Euclidean or cosine distance, we further analyze $\mathbf{z}_T^\top \mathbf{z}_0$ and $\|\mathbf{z}_0\|_2^2$ to depict a more comprehensive relationship between $\tau$ and $\mathbf{z}_T$, like the examples in Figure 3. This non-trivial trend is strictly proved in an overparameterized linear model and experimentally verified under different practical settings.

Based on these observations, we speculate that a suitable adaptation of $\mathbf{z}_T$ is beneficial when the pretrained features are not perfect for the downstream tasks. Under different experimental settings, we illustrate how to achieve better downstream performance by controlling the adaptations of $\mathbf{z}_T$ (using early-stopping HP, label smoothing during HP, or manipulating the head's capacity).

Finally, there are still many open questions and phenomena not address by this study. For example, methods to quantitatively or even adaptively analyze the discrepancy between the pretrain and downstream tasks would be very useful in knowing what kind of head probing to perform in advance. However, we believe the methods proposed in this paper can help provide a new perspective on understanding feature adaptation, which we hope will aid future work in this area.

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

Figure 5: The flow of this paper. Code is available at https://github.com/Joshua-Ren/how_to_prepare_taskhead.

## A    DECOMPOSITION OF CHANGE OF Z

**Proof of Equation (2):**

Recall $\mathbf{z} = f_{\mathbf{B}}(\mathbf{x})$, $\mathbf{q} = g_{\mathbf{z}}(\mathbf{z})$ and $\mathbf{p} = \text{Softmax}(\mathbf{q})$. Use $\mathbf{b} \in \mathbb{R}^{hd \times 1}$ to represent the vector form of $\mathbf{B} \in \mathbb{R}^{h \times d}$. The MSE loss on one sample is $\mathcal{L}_{mse}\left(\mathbf{q}^{(n)}, \mathbf{e}_{y_n}\right) = \frac{1}{2}\|\mathbf{q}^{(n)} - \mathbf{e}_{y_n}\|_2^2$. The cross-entropy loss on one sample is $\mathcal{L}_{ce}\left(\mathbf{q}^{(n)}, \mathbf{e}_{y_n}\right) = \mathcal{H}\left(\mathbf{e}_{y_n}, \mathbf{q}^{(n)}\right)$, where $\mathcal{H}(\mathbf{p}, \mathbf{q}) = -\sum_k p_k \log q_k$ is the cross entropy for categorical distribution. Using 1st Taylor expansion, we have:

$$\underbrace{\mathbf{z}_{t+1}^{(0)} - \mathbf{z}_t^{(0)}}_{h \times 1} = \underbrace{\nabla_{\mathbf{b}} \mathbf{z}_t^{(0)}}_{h \times hd} \cdot \underbrace{(\mathbf{b}_{t+1} - \mathbf{b}_t)}_{hd \times 1} + \mathcal{O}\left(\|\mathbf{b}_{t+1} - \mathbf{b}_t\|^2\right). \tag{3}$$

We then calculate $\mathbf{b}_{t+1} - \mathbf{b}_t$ assuming the parameters are updated in batch-SGD:

$$\underbrace{\mathbf{b}_{t+1} - \mathbf{b}_t}_{hd \times 1} = -\frac{\gamma}{N} a \sum_{n=1}^{N} \left( \underbrace{\nabla_{\mathbf{b}}\mathcal{L}\left(\mathbf{q}_t^{(n)}, \mathbf{e}_{y_n}\right)}_{1 \times hd} \right)^{\top} \tag{4}$$

$$= -\frac{\gamma}{N} \sum_{n=1}^{N} \left( \underbrace{\nabla_{\mathbf{q}}\mathcal{L}\left(\mathbf{q}_t^{(n)}, \mathbf{e}_{y_n}\right)}_{1 \times k} \cdot \underbrace{\nabla_{\mathbf{z}}\mathbf{q}_t^{(n)}}_{k \times h} \underbrace{\nabla_{\mathbf{b}}\mathbf{z}_t^{(n)}}_{h \times hd} \right)^{\top} \tag{5}$$

$$= -\frac{\gamma}{N} \sum_{n=1}^{N} \underbrace{\left(\nabla_{\mathbf{b}}\mathbf{z}_t^{(n)}\right)^{\top}}_{hd \times h} \underbrace{\left(\nabla_{\mathbf{z}}\mathbf{q}_t^{(n)}\right)^{\top}}_{h \times k} \underbrace{\left(\nabla_{\mathbf{q}}\mathcal{L}\left(\mathbf{q}_t^{(n)}, \mathbf{e}_{y_n}\right)\right)^{\top}}_{k \times 1} \tag{6}$$

For different loss functions, we have different expressions for the last term:

$$\nabla_{\mathbf{q}}\mathcal{L}_{mse}\left(\mathbf{q}_t^{(n)}, \mathbf{e}_{y_n}\right) = \left(\mathbf{q}_t^{(n)} - \mathbf{e}_{y_n}\right); \quad \nabla_{\mathbf{q}}\mathcal{L}_{ce}\left(\mathbf{q}_t^{(n)}, \mathbf{e}_{y_n}\right) = \left(\mathbf{p}_t^{(n)} - \mathbf{e}_{y_n}\right), \tag{7}$$

where $\mathbf{q}$ is the logits and $\mathbf{p} = \text{Softmax}(\mathbf{q})$ is the predicting probability. We see these two kinds of loss have similar expression on this term, so we only explain the cross-entropy version (as it is more common in practices) in the main content.

Using the above expressions, we can first bound the high-order term (cross-entropy version):

$$\mathcal{O}(\|\mathbf{b}_{t+1} - \mathbf{b}_t\|^2) = \mathcal{O}(\gamma^2 \|\nabla_{\mathbf{z}}\mathbf{q}_t^{(n)}\|_{op}^2 \cdot \|\nabla_{\mathbf{b}}\mathbf{z}_t^{(n)}\|_{op}^2 \cdot \|\mathbf{q}_t^{(n)} - \mathbf{e}_{y_n}\|_{op}^2) = \mathcal{O}(\gamma^2), \tag{8}$$

as long as the hyperparameters are appropriately chosen and the loss doesn't blow up (or gradient clipping is applied) in FT stage.

Finally, combining all the expressions, Equation (10) can be rewritten as:

$$
\underbrace{\mathbf{z}_{t+1}^{(0)} - \mathbf{z}_t^{(0)}}_{h \times 1} = -\frac{\gamma}{N} \sum_{n=1}^{N} \underbrace{\left( \nabla_{\mathbf{b}} \mathbf{z}_t^{(0)} \right)}_{h \times hd} \underbrace{\left( \nabla_{\mathbf{b}} \mathbf{z}_t^{(0)} \right)^{\top}}_{hd \times h} \underbrace{\left( \nabla_{\mathbf{z}} \mathbf{q}_t^{(n)} \right)^{\top}}_{h \times k} \underbrace{\left( \nabla_{\mathbf{q}} \mathcal{L}_{ce} \left( \mathbf{q}_t^{(n)}, \mathbf{e}_{y_n} \right) \right)^{\top}}_{k \times 1} + \mathcal{O}(\gamma^2) \quad (9)
$$

$$
= \frac{\gamma}{N} \sum_{n=1}^{N} \underbrace{\kappa_t^{(0,n)}}_{h \times h} \cdot \underbrace{\left( \nabla_{\mathbf{z}} \mathbf{q}_t^{(n)} \right)^{\top}}_{h \times k} \underbrace{\left( \mathbf{e}_{y_n} - \mathbf{p}_t^{(n)} \right)}_{k \times 1} + \mathcal{O}(\gamma^2), \quad (10)
$$

which is the same with Equation (2) in the main context.

**Proof of Proposition 1:**

It is easy to get $\mathbf{z}_T^{(j)} - \mathbf{z}_0^{(j)}$ by stacking the LHS of Equation (2) (or Equation (10)) under different $t$:

$$
\mathbf{z}_T^{(j)} - \mathbf{z}_0^{(j)} = \sum_{t=0}^{T-1} \frac{\gamma}{N} \sum_{n=1}^{N} \left( \kappa_t^{(j,n)} \cdot \left( \nabla_{\mathbf{z}} \mathbf{q}_t^{(n)} \right)^{\top} \cdot \left( \mathbf{e}_{y_n} - \mathbf{p}_t^{(n)} \right) \right) + \mathcal{O}(\gamma^2) \quad (11)
$$

$$
= \gamma \sum_{t=0}^{T-1} \mathbb{E}_{\mathbf{x}^{(n)}} \left[ \kappa_t^{(j,n)} \cdot \left( \nabla_{\mathbf{z}} \mathbf{q}_t^{(n)} \right)^{\top} \cdot \left( \mathbf{e}_{y_n} - \mathbf{p}_t^{(n)} \right) \right] + \mathcal{O}(\gamma^2) \quad (12)
$$

$$
= \gamma \cdot \mathbb{E}_{\mathbf{x}^{(n)}} \left[ \kappa^{(j,n)} \cdot \sum_{t=0}^{T-1} \left( \nabla_{\mathbf{z}} \mathbf{q}_t^{(n)} \right)^{\top} \cdot \left( \mathbf{e}_{y_n} - \mathbf{p}_t^{(n)} \right) \right] + \mathcal{O}(\gamma^2) \quad (13)
$$

$$
= \gamma \cdot \mathbb{E}_{\mathbf{x}^{(n)}} \left[ \kappa^{(j,n)} \cdot \sum_{t=0}^{T-1} \mathbf{v}_t^{\top} \cdot \left( \mathbf{e}_{y_n} - \mathbf{p}_t^{(n)} \right) \right] + \mathcal{O}(\gamma^2), \quad (14)
$$

where the first equation follows definition. The second equation is assuming a uniform training sample distribution, i.e., $p(\mathbf{x}) = \frac{1}{N}$. The third equation follows the slow-change NTK assumption, hence $\kappa^{(j,n)}$ no longer depends on $t$. The last equation follows the linear-head assumption, i.e., $\nabla_{\mathbf{z}} \mathbf{q}_t^{(n)} = \mathbf{v}_t$.

We cannot go further as there are three matrices (or vectors) in the expectation. Hence we instead analyze the F-norm (i.e., L2-norm for the vector $\mathbf{z}$) of the features' change.

$$\|\mathbf{z}_T^{(j)} - \mathbf{z}_0^{(j)}\|_2 = \gamma \cdot \left\| \mathbb{E}_{\mathbf{x}^{(n)}} \left[ \kappa^{(j,n)} \cdot \sum_{t=0}^{T-1} \mathbf{v}_t^{\top} \cdot \left( \mathbf{e}_{y_n} - \mathbf{p}_t^{(n)} \right) \right] \right\|_2 + \mathcal{O}(\gamma^2) \tag{15}$$

$$\leq \gamma \cdot \mathbb{E}_{\mathbf{x}^{(n)}} \left\| \kappa^{(j,n)} \cdot \sum_{t=0}^{T-1} \mathbf{v}_t^{\top} \cdot \left( \mathbf{e}_{y_n} - \mathbf{p}_t^{(n)} \right) \right\|_2 + \mathcal{O}(\gamma^2) \tag{16}$$

$$\leq \gamma \cdot \mathbb{E}_{\mathbf{x}^{(n)}} \left\| \kappa^{(j,n)} \right\|_2 \cdot \left\| \sum_{t=0}^{T-1} \mathbf{v}_t^{\top} \cdot \left( \mathbf{e}_{y_n} - \mathbf{p}_t^{(n)} \right) \right\|_2 + \mathcal{O}(\gamma^2) \tag{17}$$

$$\leq \gamma \cdot C_1 \cdot \mathbb{E}_{\mathbf{x}^{(n)}} \left\| \sum_{t=0}^{T-1} \mathbf{v}_t^{\top} \cdot \left( \mathbf{e}_{y_n} - \mathbf{p}_t^{(n)} \right) \right\|_2 + \mathcal{O}(\gamma^2) \tag{18}$$

$$= \gamma \cdot C_1 \cdot \sum_{t=0}^{T-1} \mathbb{E}_{\mathbf{x}^{(n)}} \left\| \mathbf{v}_t^{\top} \cdot \left( \mathbf{e}_{y_n} - \mathbf{p}_t^{(n)} \right) \right\|_2 + \mathcal{O}(\gamma^2) \tag{19}$$

$$\leq \gamma \cdot C_1 \cdot \sum_{t=0}^{T-1} \mathbb{E}_{\mathbf{x}^{(n)}} \left\| \mathbf{v}_t^{\top} \right\|_2 \cdot \left\| \mathbf{e}_{y_n} - \mathbf{p}_t^{(n)} \right\|_2 + \mathcal{O}(\gamma^2) \tag{20}$$

$$\leq \gamma \cdot C_1 \cdot C_2 \cdot \sum_{t=0}^{T-1} \mathbb{E}_{\mathbf{x}^{(n)}} \left\| \mathbf{e}_{y_n} - \mathbf{p}_t^{(n)} \right\|_2 + \mathcal{O}(\gamma^2) \tag{21}$$

$$\leq \gamma \cdot C_1 \cdot C_2 \cdot T \cdot \mathbb{E}_{\mathbf{x}^{(n)}} \left\| \mathbf{e}_{y_n} - \mathbf{p}_0^{(n)} \right\|_2 + \mathcal{O}(\gamma^2). \tag{22}$$

Here the first equation is by definition. The second inequality follows triangle inequality. The third inequality follows Cauchy-Schwarz inequality. The forth inequality is assuming the F-norm of NTK is bounded by $C_1$. The fifth equation is sweeping the summation order. The sixth inequality also follows Cauchy-Schwarz inequality. The seventh inequality assumes the norm of the last layer is bounded by $C_2$. The eighth inequality is assuming a stable learning process where the gap between $\mathbf{e}_{y_n}$ and $\mathbf{p}_0^{(n)}$ keeps decreasing during training.

Finally, by taking expectation of all the input samples, we can have:

$$\mathbb{E}_{\mathbf{x}^{(j)}} \|\mathbf{z}_T^{(j)} - \mathbf{z}_0^{(j)}\|_2 \leq c \cdot \mathbb{E}_{\mathbf{x}^{(n)}} \|\mathbf{e}_{y_n} - \mathbf{p}_0^{(n)}\|, \tag{23}$$

where $c = \gamma \cdot C_1 \cdot C_2 \cdot T$ is a constant and $\mathbb{E}_{\mathbf{x}^{(n)}} \|\mathbf{e}_{y_n} - \mathbf{p}_0^{(n)}\|$ is the AIE term ($E_{aie}$) defined in Proposition 1.

**Verification of Slow Kernel Change Assumption:**

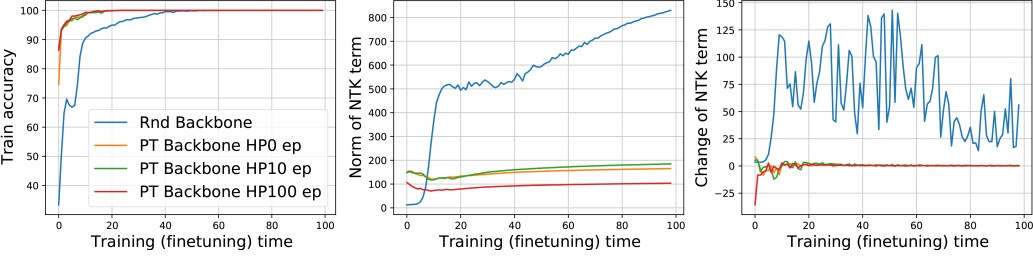

Figure 6: In finetuning, the NTK of the backbone adapts quite slow if the model is pretrained.

We decompose the one-step dynamics of $\mathbf{z}_t^{(j)}$ in Equation (2) into three parts, which represent kernel, direction, and energy respectively. In our analysis, we have a mild assumption that the kernel of the backbone, i.e., $\kappa_t^{(j,n)} = \left( \nabla_{\mathbf{B}} \mathbf{z}_t^{(j)} \right) \left( \nabla_{\mathbf{B}} \mathbf{z}_t^{(n)} \right)^{\top} \in \mathbb{R}^{h \times h}$, changes slow during finetuning when the learning rate is small and the training is stable (the loss converges). Although there are several

works supporting this assumption (Yang & Hu, 2020; Geiger et al., 2020), to make the paper more self-contained, we directly observe how this term adapts during pretraining in Figure 6.

We measure the change of this term by calculating $k_{norm} = \|\kappa_t^{(j,n)}\|_F$ and $k_{gap} = \|\kappa_{t+1}^{(j,n)} - \kappa_t^{(j,n)}\|_F$ for each $t$ during finetuning. As computing the empirical NTK on the whole dataset requires huge memory, we randomly sample 50 $\mathbf{x}$ as our "probing samples", computing $k_{norm}$ and $k_{gap}$ on 2,500 different $(\mathbf{x}^{(j)}, \mathbf{x}^{(n)})$ pairs, and then report their mean values at each $t$. To verify our assumption, we compare four different settings:

- In Rnd Backbone, we randomly initialize the whole network and train it on the downstream dataset;
- In PT Backbone HP$\tau$ ep, we copy the pretrained backbone, attach a randomly initialized head, and then HP for $\tau$ epochs.

In the first panel in Figure 6, we see the training accuracy of all these settings converge to 100% (they also have similar validation accuracy). In the second and third panel, we plot the change of $k_{norm}$ and $k_{gap}$ respectively. It is obvious that compared with using random backbone, the NTK of a pretrained backbone indeed changes slow during the finetuning stage.

## B  EXPERIMENTAL SETTINGS

### B.1  TOY CASE

The experiments of the toy setting appear in Figure 2 and Figure 3. In the toy setting, we first pretrain a 4+1 (4 layers of backbone and 1 layer of task head) layers MLP on the full MNIST dataset (LeCun, 1998), with learning rate $10^{-3}$, with cosine scheduler until convergence. The hidden width is 128 and relu activation is applied for each layer in the backbone. In the downstream task, we only consider the first three classes and randomly select 1000 samples for each class (i.e., 3*1000 samples in total). We also apply random rotation and center crop augmentations to simulate the distribution shift in downstream task, like the right panel in Figure 7. The downstream linear task head is then a 128*3 matrix (weights) and a 128*1 vector (bias). To ensure a fine-grained observation of feature's adaptation, we use a small constant learning rate ($10^{-4}$) in all the experiments. The FT epoch is 50, and the HP epoch ranges from 0 to 50,000 (50,000 is simulating the $\mathbf{v}_{hp}^\infty$ case, usually train 0 to 1,024 epochs).

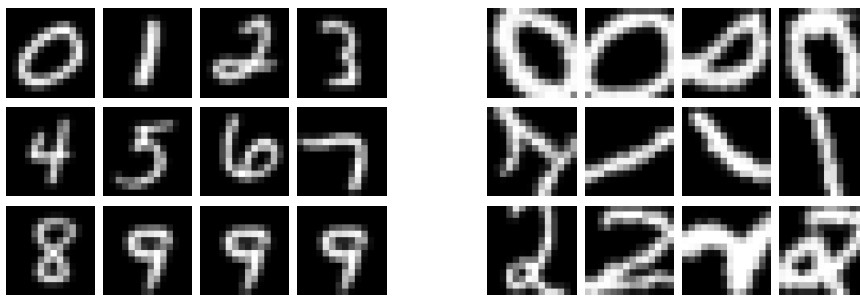

(a) Pretrain dataset.                    (b) Downstream (HP/FT) dataset.

Figure 7: Dataset for toy experiments. We use full MNIST dataset to pretrain the network. The downstream dataset is a subset of MNIST (only 0, 1 and 2) applying random rotation and zoom-in.

### B.2  PRACTICAL SETTINGS – CLASSIFICATION

Besides the simple MNIST and MLP, we also verify our analysis in many real settings. To demonstrate the generality of our findings, we conduct experiments on image classification, image seg-

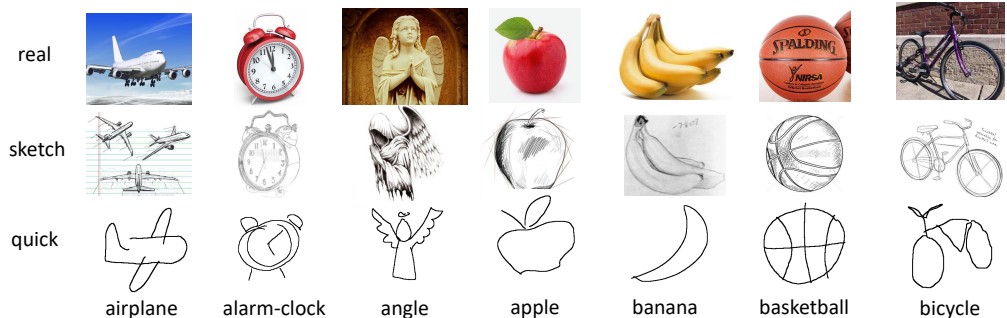

Figure 8: DomainNet dataset. Only three domains are used: 'real' is similar to IN1K, 'sketch' is less sensitive to color and background, 'quick' only contains some lines, which is quite different from IN1K.

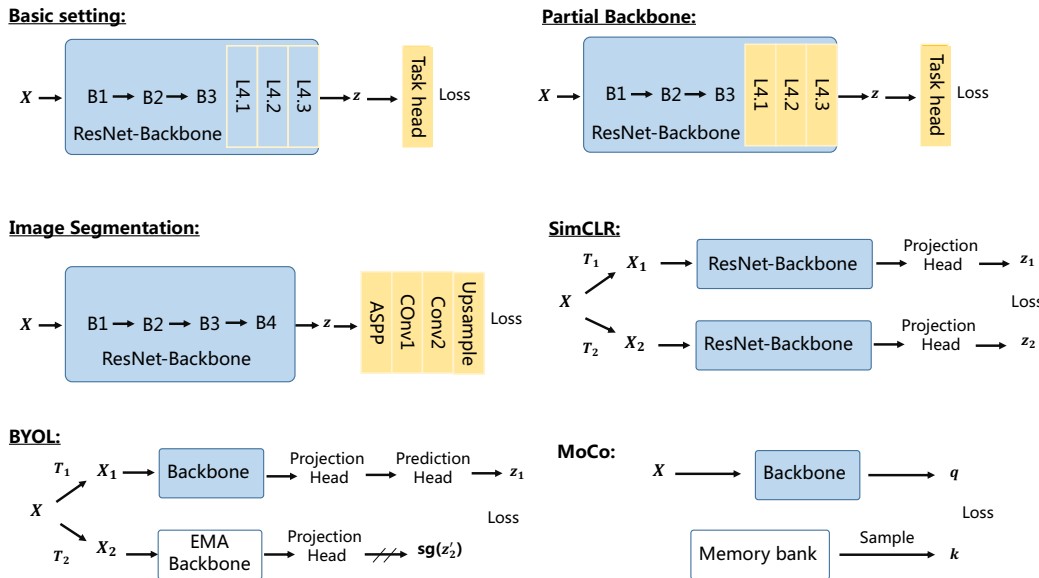

Figure 9: Introduction of models used in this paper.

mentation, and molecular graph classification tasks. The qualitative trend is quite consistent across different tasks and networks.

**Model Structure:**

Specifically, we consider ResNet18/34/50 for the image classification task. They all have 4 blocks, each containing 3 layers, as illustrated in Figure 9. For the linear task head (or an MLP head), we directly concatenate a linear layer (or an MLP layer) to the backbone. For the partial backbone method, several layers in the last block (e.g., L4.3, L4.2, L4.1) might be reinitialized and merged to the task head. We also provide what is the backbone (all the blue squares) when self-supervised-learning is considered, e.g., SimCLR, BYOL, and MoCo. In short, after pre-training, the backbones under different PT tasks might have the same structure.

**Pretraining tasks:**

We pretrain the ResNet on CIFAR10 using common recipe ($10^{-2}$ learning rate with a cosine scheduler (He et al., 2019), $5 * 10^{-4}$ weight decay, simple augmentations like random flipping and cropping, etc.). For the model pretrained on ImageNet, we directly download the checkpoints from open-source implementations. Due to time and computing resource limitations, for vision tasks, we only consider MoCo (He et al., 2020), Byol (Grill et al., 2020), SimCLR (Chen et al., 2020a), and

supervised classification tasks for pretraining. For the graph dataset case, we consider a fundamental 5-layers graph convolutional network (GCN, (Kipf & Welling, 2017)), the hyper-parameters and other detailed settings can be found in our github repo. The datasets applied in this paper are listed in Table 4. Note that the backbone of image segmentation is the same as that in image classification. The segmentation head is more complex, which will be discussed later.

**Hyper-parameters for HP and FT:**

In this paper, experiments with the same setting in different figures or tables share the same set of hyper-parameters. Generally, for all the experiments, the batch size is 128, hidden layer width is 256 (in the MLP head case). The input image size for CIFAR pretrained model is $32 \times 32$ while that for the IN1K pretrained model is $224 \times 224$. Hence in the corresponding experiments, we will first resize the input samples and then apply random cropping and random flipping augmentations. For the HP phase, the $\tau^*$ is selected based on the validation performance, but the detailed learning rate and maximum $\tau$ might be different under different settings (as the dataset size are different). For the FT phase, we use a standard SGD with momentum ($\beta = 0.9$). The default learning rate is $10^{-3}$ and a cosine scheduler is applied (the maximum FT epoch also varies for different downstream datasets). Note that we will early stop the FT process, hence the maximum FT epoch doesn't influence the reported performance a lot. For those dataset-dependent hyper-parameters, we summarize them as:

- For the insufficient pretrained backbone cases, i.e., Res50-C10(Sup), Res18-C10(Sup), in Figure 4, Table 1 and other related experiments in the appendix, we set $\tau \in [0, 200]$, HP learning rate is $3 * 10^{-2}$, maximum FT epochs is 200 (usually converge less than 100 epochs), with usually a larger initial FT learning rate (e.g., $3 * 10^{-3}$);

- For the IN1K pretrained backbone (no matter by supervisory training, MoCo, Byol or SimCLR), which can be found in Figure 1, 2, 4, Table 1 and many figures in the appendix, we usually set $\tau \in [0, 200]$, HP learning rate as $10^{-2}$, maximum FT epochs as 200, and a relative small FT learning rate (e.g., $3 * 10^{-4}$);

- For any experiments whose downstream tasks are Flowers, Cars (the dataset is small while number of classes is large), we consider to increase the HP learning rate to $5 * 10^{-2}$, and the maximum FT epochs to 1000 (usually converge less than 200 epochs);

- For any experiments whose downstream task is CIFAR (the dataset is bigger than STL), we consider to increase the HP learning rate to $5 * 10^{-3}$, and the maximum FT epochs to 100 (usually converge less than 50 epochs);

In summary, a general principle for these hyper-parameter selections is that the maximum $\tau$ should make the HP training accuracy converge (increase slowly for several consecutive epochs), while the finetuning epochs should make the training accuracy converge when $\tau = 0$. For example, if the downstream task is CIFAR10, which has 50,000 training samples and only 10 classes, finetune 20 epochs is enough. But CIFAR100, which also has 50,000 training samples but with more classes, 50 finetuning epochs are required (we set the maximum FT epochs as 100 for both CIFAR10 and CIFAR100). For Flowers102, which only has 6,149 training samples but 102 classes, we set the maximum FT epochs as 1,000 epochs. Anyway, we find the proposed trend is quite robust to these hyper-parameters. The detailed settings can be found in our code base.

### B.3 PRACTICAL SETTINGS – IMAGE SEGMENTATION

In addition to the image classification tasks, we also conduct experiments on image segmentation tasks to verify the robustness of our analysis on various tasks. An image segmentation task is a task where a model is trained to make class predictions at each pixel of an image given an input image. We train and test segmentation models on PASCAL VOC (Everingham et al., 2015) dataset, one of the most popular image segmentation datasets. An example of input image and segmentation label are provided in Figure 10.

**Model Structure:**

For image segmentation tasks, we employ DeeplabV3 (Chen et al., 2017a) with ResNet50 as a backbone. As with the classification tasks, ResNet50 consists of 4 blocks, each of which contains 3 layers. But unlike the classification tasks, kernel strides for convolutional layers are adjust to have

|              | # train   | # test   | # class    | Comments |
|--------------|-----------|----------|------------|----------|
| **MNIST**        | 60,000    | 10,000   | 10         | Toy downstream use a subset, as in Figure 7. |
| **STL10**        | 5,000     | 8,000    | 10         | Coates et al. (2011) |
| **CIFAR10**      | 50,000    | 10,000   | 10         | Krizhevsky et al. (2009) |
| **CIFAR100**     | 50,000    | 10,000   | 100        | Krizhevsky et al. (2009) |
| **Flowers102**   | 6,149     | 2,040    | 102        | Nilsback & Zisserman (2008) |
| **StanfordCars** | 8,144     | 8,041    | 196        | Krause et al. (2013) |
| **Dom-real**     | <20,000   | <4,000   | 200 (345)  | A subset, original 345 classes and more samples. |
| **Dom-sketch**   | <20,000   | <4,000   | 200 (345)  | Less sensitive to color and texture. |
| **Dom-quick**    | <20,000   | <4,000   | 200 (345)  | Only contains simple curves. (Peng et al., 2019) |
| **ImageNet-1K**  | 1,281,167 | –        | 1000       | Deng et al. (2009) |
| ogbg-moltox21 | 6,272     | 1,742    | 12         | Wu et al. (2018) |
| ogbg-molhiv   | 32,901    | 8,226    | regression | Hu et al. (2020) |
| ogbg-molpcba  | 437,929   | –        | 128        | Hu et al. (2020) |

Table 4: Datasets (vision and molecular graph) used in experiments.

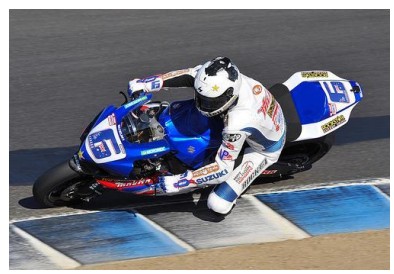

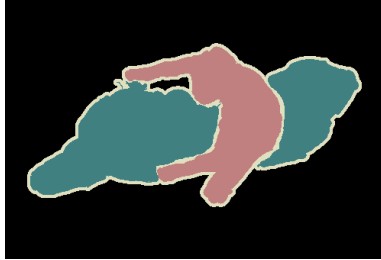

(a) An input image.                    (b) A segmentation label.

Figure 10: An example of an image and label pair in PASCAL VOC dataset.

the higher spatial resolution for the output features of the backbone; the spatial resolution of the features is $7 \times 7$ for the classification tasks whereas it is $17 \times 17$ for the segmentation tasks.

The features extracted from the ResNet50 backbone are passed to a segmentation head, which consists of Atrous Spatial Pyramid Pooling (ASPP) (Chen et al., 2017b) layers followed by a couple of convolutional layers. ASPP is a stack of five regular and dilated convolutional layers which provide the features with various size of receptive fields. The spatial resolution of the outputs of the segmentation head stays in $17 \times 17$, which are then upsampled to the original input image size through bilinear interpolation. Here, we use the whole segmentation head as the task head.

**Hyper-parameters for HP and FT:** The ResNet50 backbone is pretrained on ImageNet dataset. We set $\tau \in [0, 200]$, HP learning rate as $0.3$, maximum FT epochs as $200$ (usually converge less than 50 epochs). Also, we use batch size of 16, and a SGD optimizer with momention ($\beta = 0.9$) but without weight decay nor learning rate scheduler.

## C  MORE EXPERIMENTS

### C.1  VERIFICATION ON THE CHANGE OF Z

In Equation (2), we decompose the learning dynamics of $\mathbf{z}_t$ into three parts: relatively stable kernel, direction, and energy. Based on this, we can expect a larger adaptation when the initial energy is large. Furthermore, to get a more precise description of $\mathbf{z}_t$'s adaptation, we analyze a simple overparameterized linear model in Appendix D. We show that as the initial prediction gap (i.e., energy) increases, the resulting $\mathbf{z}_t$ will first be stretched with a small direction change (compared with the original $\mathbf{z}_0$). If the initial energy keeps increasing, the resulting $\mathbf{z}_t$ might be more and more dissimilar to the original one. We use four distance related quantity to describe this trend, and verify it under many different settings. In short, the trends are: 1.) more energy leads to larger $\|\mathbf{z}_t - \mathbf{z}_0\|_2^2$ and smaller $\cos(\mathbf{z}_t, \mathbf{z}_0)$; 2.) $\mathbf{z}_t^\top \mathbf{z}_0$ and $\|\mathbf{z}_t\|_2^2$ has a quadratic shape when energy decrease. Figure 11 provides an example of how $\mathbf{z}_t$ changes when different $\tau$ is chosen.

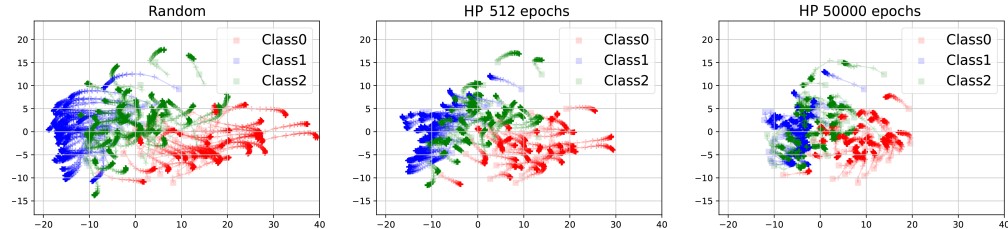

Figure 11: Change of $\mathbf{z}$ during FT under a toy setting. Using transparent square markers, we first draw the 2-D projections (the first two components in PCA) of 100 randomly selected $\mathbf{z}_0^{(n)}$. Then keeping the eigen-directions of this PCA, we project $\mathbf{z}_t^{(n)}$ of different $t$ and encode $t$ by transparency. The converged $\mathbf{z}$ is represented by a cross marker with full transparency. It is clear that FT using $\mathbf{v}_{rnd}$ makes $\mathbf{z}$ change a lot, while $\mathbf{v}_{hp}^{50000}$ doesn't change $\mathbf{z}$ too much.

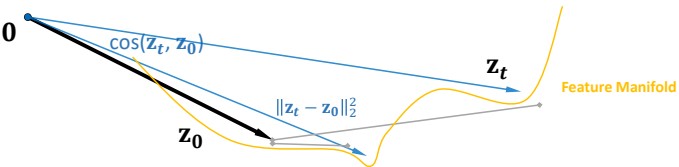

Figure 12: Explanation of why small change of $\mathbf{z}_t^\top \mathbf{z}_0$, $cos(\mathbf{z}_t, \mathbf{z}_0)$ can influence a lot.

Before diving deep into these examples, we provide some general commands. First, we do not expect the experimental results to align perfectly with the examples in Figure 3, because too many designs (like dropout, data augmentation, SGD noise, locking in strange local minima, learning rate schedule, etc.) can influence the shape of the curve, hence we believe observing specific trends in many cases is already a strong support for our analysis. Second, one might be curious about why in some cases, the ranges between $\mathbf{z}_t^\top \mathbf{z}_0$ and $cos(\mathbf{z}_t, \mathbf{z}_0)$ are so small. Hence in Figure 12, we provide an illustration of why this happens: as the dot product measures two long vectors (remember they have the same origin), a small change of $\mathbf{z}_t^\top \mathbf{z}_0$ can make a big difference. A large range of $\mathbf{z}_t^\top \mathbf{z}_0$ and $cos(\mathbf{z}_t, \mathbf{z}_0)$ is a sign that $\mathbf{z}_t$ already moves to another basin, which is the strong-case in Figure 3. Third, in Appendix D, the change of $\mathbf{z}$ is linked to $\mathbf{q}_0 - \mathbf{e}_y$ under an MSE loss, while in these experiments, we use $\tau$ as our x-axis and consider a cross-entropy loss. Combining with the NTK approximation applied in this ideal model, the experimental trends might not be exactly the same with the theoretical ones.

Then, in Figure 13, Figure 14 and Figure 15, we demonstrate the trend when the model is pretrained on different datasets, on different tasks, and when the downstream tasks are different, respectively. Besides the general trends, i.e., decreasing $\|\mathbf{z}_t - \mathbf{z}_0\|_2^2$, increasing $cos(\mathbf{z}_t, \mathbf{z}_0)$, quadratic $\mathbf{z}_t^\top \mathbf{z}_0$ and $\|\mathbf{z}_t\|_2^2$, we can observe another interesting phenomenon when the model is pretrained using different tasks. See Figure 14, in Byol and SimCLR, sometimes we might get a very big $\mathbf{z}_t^\top \mathbf{z}_0$ (together with a very small $cos(\mathbf{z}_t, \mathbf{z}_0)$), which makes it hard to observe any quadratic trend in some metrics. But in the supervised pretraining case (no matter what dataset we use in pretrain and downstream tasks), we never observe such a phenomenon – the $\|\mathbf{z}_t - \mathbf{z}_0\|_2^2$ always changes in a relatively small range. We speculate that in the first several updates during head probing, the features first adapt to the downstream task (i.e., classification), then gradually adapt to the downstream dataset distribution (e.g., STL, Flowers, etc.). So a randomly initialized head might influence more when the pretrain task is different from the downstream one. The detailed mechanism of adaptation of task and data distribution might be more complex than we expect, so we left this for the future work.

## C.2 INFLUENCE OF THE BACKBONE AND THE TASK-HEAD CAPACITY

In Section 4.3, we mention a case where the task head is more complex. For example, the head can be a two-layers MLP rather than one linear layer, or we can only copy part of the pretrained model as our backbone. However, these methods and tricks are quite complex and might not necessarily enhance the performance, hence we put some results and discussions here. Remember that our

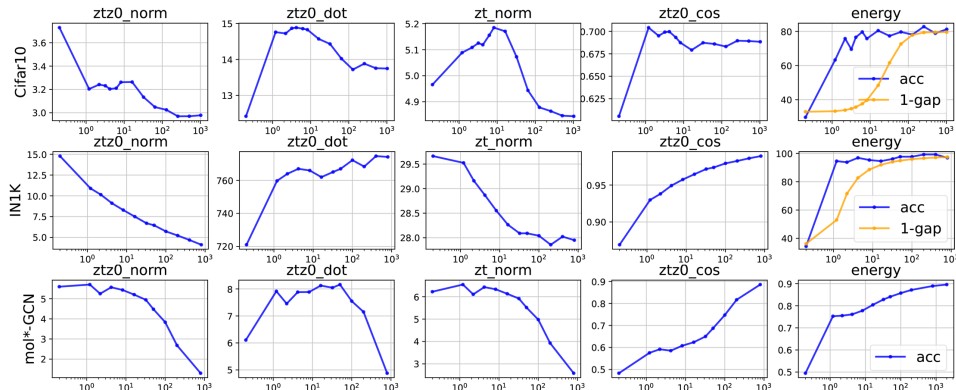

Figure 13: Adaptation of $\mathbf{z}_t$ when model is pretrained on different datasets.

analysis of how energy influences the features adaptation is still valid in these cases, as illustrated by the distance related metrics in the tables.

From Table 5, 6, 7, and 8, we can draw the following three conclusions. First, the two-layers MLP design will surely increase the training accuracy after head probing, and hence make the features adapt less (but this might not enhance the downstream performance). Second, we compare the downstream performance when some layers of the backbone are reinitialized as part of the task head. The title "+L4.3" means we take the last layer (i.e., the 3-rd layer) of the last block (i.e., the 4-th block) in a ResNet50 out, and treat it as part of the task head. The title "+L4.2" and "+L4.1" means we continue treating the 2-nd or the 1-st layer as the backbone. Under such a setting, the "+L4.1" case will inherit the least amount of information from the pretrained model, and at the same time, have the biggest task head. For the results in these tables, we train the head until convergence in head probing phase, then fine tune the whole network together. Hence we see the "HP-train-acc" value all increase to roughly 100%. However, the best setting differs when different downstream datasets are considered. When the pretrained features are good (i.e., trained using IN1K), we see the Domain-real dataset needs less adaptation and usually performs the best in the baseline case (i.e., copy all parameters from the pre-trained model). But Domain-quick dataset, which only contains some black-and-white lines, prefers the "+L4.1" setting. In other words, preserving the features captured by the earlier layers of the pretrained model is beneficial. When the pretrained features are bad, like the CIFAR-pretraining case in Table 7, all these datasets prefer throwing away the later layers.

In summary, although the correlation between features adaptation and downstream task performance under different settings is quite complicated, our analysis of energy can still explain some phenomena well.

| | | Real | Sketch | Quick | | Real | Sketch | Quick |
|---|---|---|---|---|---|---|---|---|
| Linear-head Baseline | HP-train-acc | 98.994 | 63.369 | 48.555 | 2-MLP Head | 100 | 93.447 | 63.75 |
| | $1 - \cos(\mathbf{z}_t, \mathbf{z}_0)$ | 0.0315 | 0.2275 | 0.3506 | | 0.0149 | 0.2032 | 0.3375 |
| | $\|\mathbf{z}_t - \mathbf{z}_0\|^2$ | 17.868 | 22.246 | 27.954 | | 17.694 | 21.171 | 27.634 |
| | val-acc | **54.839** | **37.789** | 61.416 | | 53.856 | 35.446 | 59.199 |
| +L4.3 | val-acc | 53.629 | 35.966 | **61.568** | HP-train-acc | 99.492 | 99.072 | 93.975 |
| +L4.2 | val-acc | 53.226 | 36.69 | 61.391 | HP-train-acc | 99.033 | 99.336 | 98.193 |
| +L4.1 | val-acc | 53.856 | 35.446 | 59.199 | HP-train-acc | 99.863 | 93.447 | 63.75 |

Table 5: Results on a ResNet50 pertrained on Domain-Real. The pretrained validation accuracy is only 54.612 while the training accuracy is 100. Maybe because the DomainNet-Real dataset only contains less than 20k samples.

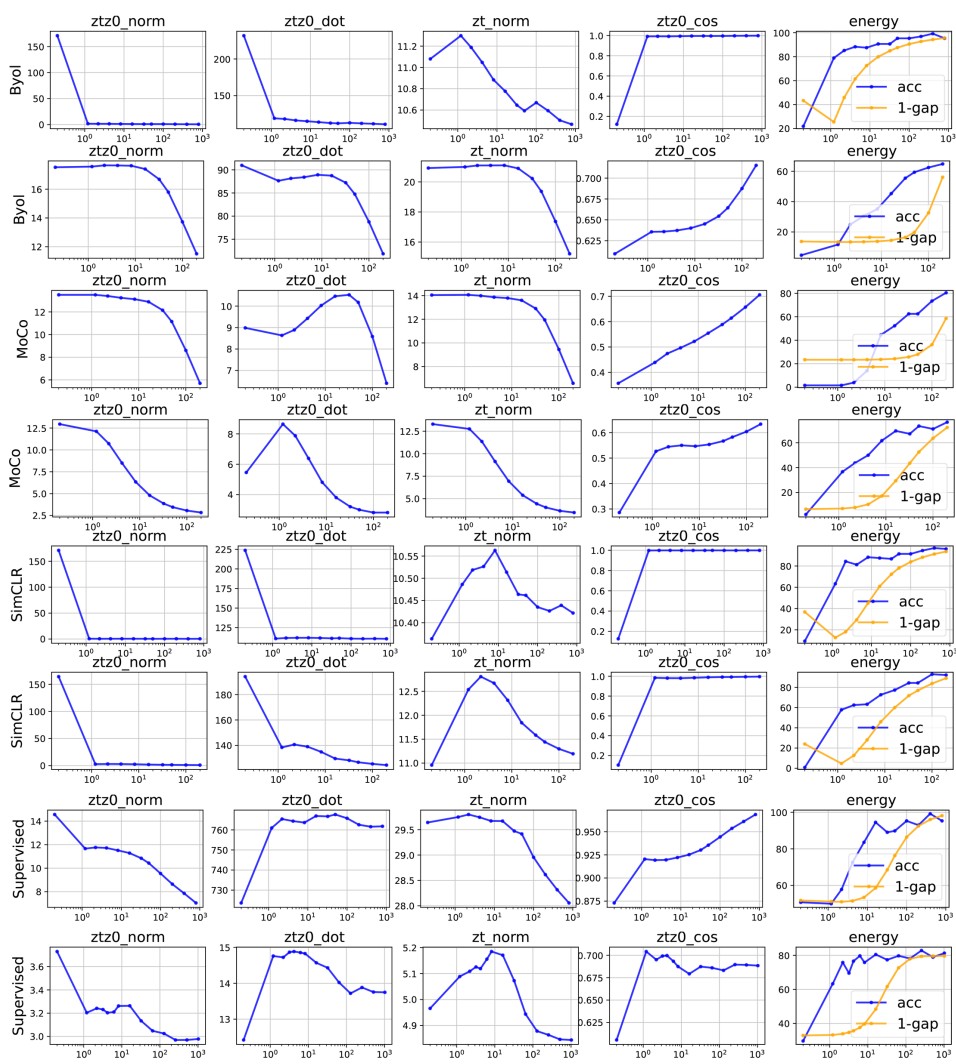

Figure 14: Adaptation of $\mathbf{z}_t$ when model is pretrained using different tasks.

| | | Real | Sketch | Quick | | Real | Sketch | Quick |
|---|---|---|---|---|---|---|---|---|
| | HP-train-acc | 60.039 | 39.092 | 51.855 | | 88.965 | 79.58 | 76.445 |
| Linear-head | $1 - \cos(\mathbf{z}_t, \mathbf{z}_0)$ | 0.3492 | 0.4181 | 0.5034 | 2-MLP | 0.3296 | 0.3672 | 0.4923 |
| Baseline | $\|\mathbf{z}_t - \mathbf{z}_0\|^2$ | 9.963 | 9.72 | 9.287 | Head | 9.104 | 9.209 | 9.874 |
| | val-acc | 58.846 | 45.023 | 61.593 | | 58.77 | 43.171 | 60.307 |
| +L4.3 | val-acc | 59.929 | 42.882 | 59.602 | HP-train-acc | 100 | 100 | 98.252 |
| +L4.2 | val-acc | 60.837 | 44.734 | 61.542 | HP-train-acc | 100 | 100 | 100 |
| +L4.1 | val-acc | **61.164** | **45.775** | **62.626** | HP-train-acc | 100 | 100 | 100 |

Table 6: Results on a ResNet34 pertrained on CIFAR-100 in a classification task.

# D   ANALYZE LINEAR OVERPARAMETERIZATION PROBLEM

## D.1   FORMALIZE THE CHANGE OF REPRESENTATIONS

Section 4 provides some intuitive explanations of how feature extractor $f(\mathbf{x}; \mathbf{B})$ changes given different $g(\mathbf{z}; \mathbf{v})$, which are well supported by the experimental results. To provide more insights, we

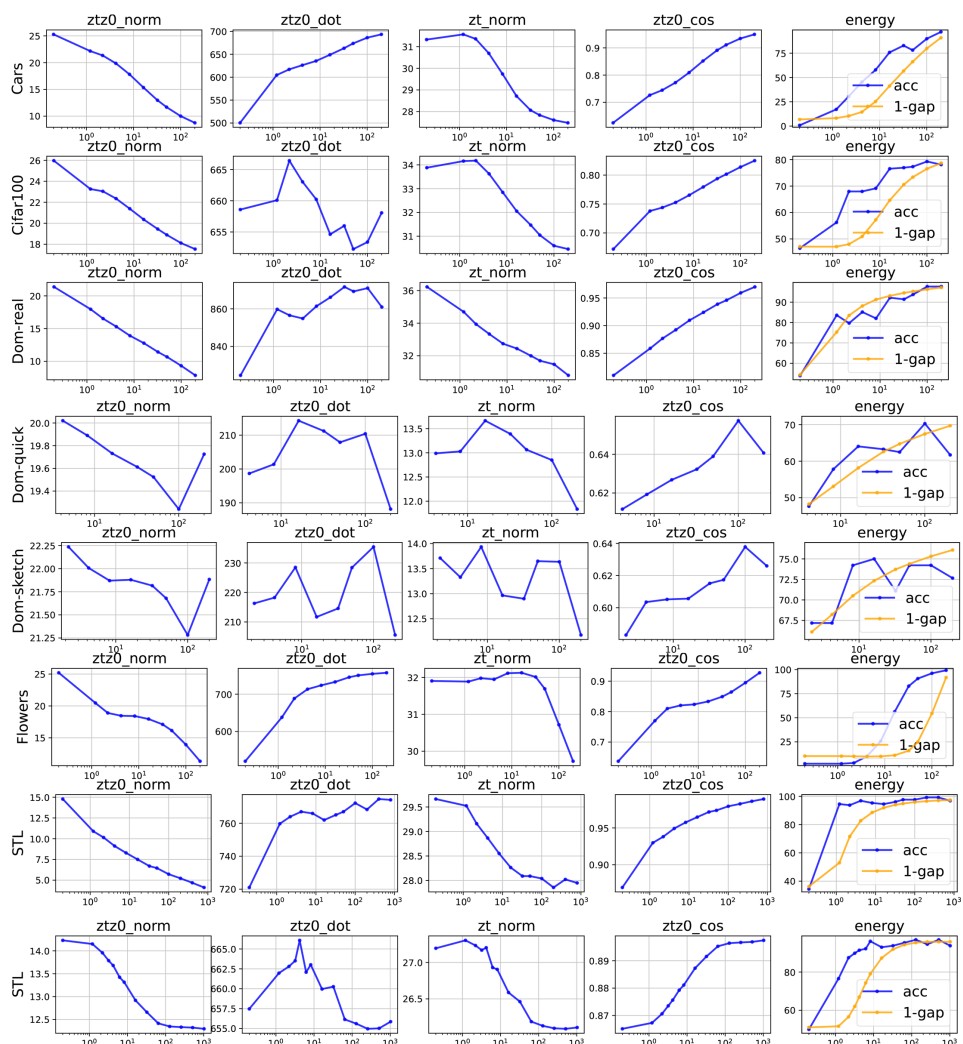

Figure 15: Adaptation of $\mathbf{z}_t$ when downstream tasks are different.

| | | Real | Sketch | Quick | | Real | Sketch | Quick |
|---|---|---|---|---|---|---|---|---|
| Linear-head Baseline | HP-train-acc | 96.875 | 85.938 | 64.063 | 2-MLP Head | 96.671 | 98.438 | 84.375 |
| | $1 - \cos(\mathbf{z}_t, \mathbf{z}_0)$ | 0.1384 | 0.2807 | 0.4432 | | 0.1544 | 0.2661 | 0.4147 |
| | $\|\mathbf{z}_t - \mathbf{z}_0\|^2$ | 16.569 | 18.005 | 21.869 | | 16.843 | 17.672 | 21.389 |
| | val-acc | **85.685** | **72.51** | **69.531** | | 85.031 | 69.651 | 66.683 |
| +L4.3 | val-acc | 84.929 | 69.734 | 68.422 | HP-train-acc | 100 | 100 | 95.232 |
| +L4.2 | val-acc | 83.342 | 68.547 | 68.7 | HP-train-acc | 100 | 100 | 98.087 |
| +L4.1 | val-acc | 81.074 | 69.155 | 68.246 | HP-train-acc | 100 | 100 | 98.842 |

Table 7: Results on a ResNet50 pertrained on ImageNet-1K in a classification task.

analyze the change of features (i.e., $\mathbf{z}$) in a simplified overparameterization problem, i.e., the one provided in Section 3 and in Kumar et al. (2022). Under some mild assumptions and approximations, we provide an overview of how the norm and direction of $\mathbf{z}$ changes under different choice of $\mathbf{v}_0$. We show that the $\mathbf{v}_0$ satisfying $\mathbf{q}_0 = Y$ or $\mathbf{q}_0 = \frac{1}{2}Y$ are two critical points in feature adaptation.

We first rewrite Equation (1) in a non-matrix form:

| | | Real | Sketch | Quick | | Real | Sketch | Quick |
|---|---|---|---|---|---|---|---|---|
| | HP-train-acc | 92.676 | 78.213 | 64.307 | | 99.121 | 94.883 | 82.354 |
| Linear-head | $1 - \cos(\mathbf{z}_t, \mathbf{z}_0)$ | 0.1269 | 0.1422 | 0.1934 | 2-MLP | 0.1098 | 0.1104 | 0.1522 |
| Baseline | $\|\mathbf{z}_t - \mathbf{z}_0\|^2$ | 5.247 | 4.752 | 4.361 | Head | 4.942 | 4.206 | 3.868 |
| | val-acc | 78.075 | 61.545 | 59.703 | | 78.453 | 63.773 | 63.609 |
| +L4.3 | val-acc | **78.528** | **65.683** | 67.011 | HP-train-acc | 100 | 100 | 97.666 |
| +L4.2 | val-acc | 78.427 | 65.336 | 67.087 | HP-train-acc | 100 | 100 | 100 |
| +L4.1 | val-acc | 76.689 | 65.017 | **68.07** | HP-train-acc | 100 | 100 | 100 |

Table 8: Results on a ResNet50 pertrained on ImageNet-1K in a SimCLR task.

$$\mathcal{L}_{\mathbf{B},\mathbf{v}} = \frac{1}{N} \sum_{n=1}^{N} \frac{1}{2} \|\mathbf{v}^\top \mathbf{B} \mathbf{x}^{(n)} - y\|_2^2, \tag{24}$$

where $y \in \mathbb{R}$ as we are considering a regression problem (or a classification problem using MSE loss). We use the subscript to represent the time step, e.g., $\mathbf{q}_0$, $\mathbf{z}_0$, $\mathbf{v}_0$ and $\mathbf{B}_0$ are output, feature, head parameters and backbone parameters before finetuning. Similarly, $\mathbf{q}_t$, $\mathbf{z}_t$, $\mathbf{v}_t$ and $\mathbf{B}_t$ are the corresponding values after finetuned $t$ steps. Note that $\mathbf{q}(X)$ and $\mathbf{z}(X)$ are functions of $N$ input samples $X \in \mathbb{R}^{N \times d}$, but we omit it for simplicity. We use lowercase letters to represent the $n$-th element of the vector. Before discussing a specific element, we will clarify whether we are discussing the initialized case or the finetuned case. For example, specifying $\mathbf{q}_t$, we use $q_n = f(\mathbf{x}_n) = (f(X))_n$ to represent the prediction of $\mathbf{x}_n$ after finetuning.

Remember the goal of this paper is finding a suitable way to select task head, i.e., $\mathbf{v}_0$, given the pretrained feature extractor, i.e., $\mathbf{B}_0$. Depending on the discrepancy between the pretraining task and the downstream task, we might expect $\mathbf{z}_t$ change differently after finetuning. In other words, we care about the expected change of $\mathbf{z}_t$ compared to $\mathbf{z}_0$, i.e., $\mathbb{E}_{\mathbf{x}\sim\mathcal{D}}[d(\mathbf{z}_t, \mathbf{z}_0)]$, where $\mathcal{D}$ is the data distribution of the downstream task. Depending on what distance measurement (i.e., $d(\cdot, \cdot)$) we choose, there are three different metrics:

- Euclidean: $\bar{d}_{euc} \triangleq \mathbb{E}_{\mathbf{x}\sim\mathcal{D}}\left[\|\mathbf{z}_t - \mathbf{z}_0\|_2^2\right]$,
- Dot product: $\bar{d}_{dot} \triangleq \mathbb{E}_{\mathbf{x}\sim\mathcal{D}}\left[\mathbf{z}_t^\top \mathbf{z}_0\right]$,
- Cosine: $\bar{d}_{\cos} \triangleq \mathbb{E}_{\mathbf{x}\sim\mathcal{D}}\left[\frac{\mathbf{z}_t^\top \mathbf{z}_0}{\|\mathbf{z}_t\|_2 \cdot \|\mathbf{z}_0\|_2}\right]$.

We start from the Euclidean case:

$$
\begin{aligned}
\bar{d}_{euc} &= \mathbb{E}_{\mathbf{x}\sim\mathcal{D}}\left[(\mathbf{z}_t - \mathbf{z}_0)^\top(\mathbf{z}_t - \mathbf{z}_0)\right] \\
&= \mathbb{E}_{\mathbf{x}\sim\mathcal{D}}\left[\mathbf{z}_t^\top \mathbf{z}_t - \mathbf{z}_t^\top \mathbf{z}_0 - \mathbf{z}_0^\top \mathbf{z}_t + \mathbf{z}_0^\top \mathbf{z}_0\right] \\
&= \mathbb{E}_{\mathbf{x}\sim\mathcal{D}}\left[\mathbf{z}_t^\top \mathbf{z}_t - 2\mathbf{z}_0^\top \mathbf{z}_t + \mathbf{z}_0^\top \mathbf{z}_0\right] \\
&= \mathbb{E}_{\mathbf{x}\sim\mathcal{D}}\left[\mathbf{x}^\top \mathbf{B}_t^\top \mathbf{B}_t \mathbf{x} - 2\mathbf{x}^\top \mathbf{B}_0^\top \mathbf{B}_t \mathbf{x} + \mathbf{x}^\top \mathbf{B}_0^\top \mathbf{B}_0 \mathbf{x}\right] \\
&= \mathbb{E}_{\mathbf{x}\sim\mathcal{D}}\left[tr(\mathbf{x}^\top(\mathbf{B}_t^\top \mathbf{B}_t - 2\mathbf{B}_0^\top \mathbf{B}_t + \mathbf{B}_0^\top \mathbf{B}_0)\mathbf{x})\right] \\
&= \mathbb{E}_{\mathbf{x}\sim\mathcal{D}}\left[tr((\mathbf{B}_t^\top \mathbf{B}_t - 2\mathbf{B}_0^\top \mathbf{B}_t + \mathbf{B}_0^\top \mathbf{B}_0)\mathbf{x}\mathbf{x}^\top)\right] \\
&= \mathbb{E}_{\mathbf{x}\sim\mathcal{D}}\left[tr((\mathbf{B}_t^\top \mathbf{B}_t - 2\mathbf{B}_0^\top \mathbf{B}_t + \mathbf{B}_0^\top \mathbf{B}_0)\mathbf{x}\mathbf{x}^\top)\right] \\
&\leq \mathbb{E}_{\mathbf{x}\sim\mathcal{D}}\left[tr(\mathbf{B}_t^\top \mathbf{B}_t - 2\mathbf{B}_0^\top \mathbf{B}_t + \mathbf{B}_0^\top \mathbf{B}_0) \cdot M\right] \\
&= \left[tr(\mathbf{B}_t^\top \mathbf{B}_t) - 2tr(\mathbf{B}_0^\top \mathbf{B}_t) + tr(\mathbf{B}_0^\top \mathbf{B}_0)\right] \cdot \mathbb{E}_{\mathbf{x}\sim\mathcal{D}}[M] \tag{25}
\end{aligned}
$$

where $M \triangleq tr(\mathbf{x}\mathbf{x}^\top)$ or $M \triangleq \|\mathbf{x}\mathbf{x}^\top\|_{op})$.

Similarly, we can get the expressions of $\bar{d}_{dot}$ and $\|\mathbf{z}_t\|_2^2$, which are building blocks of $\bar{d}_{\cos}$:

$$\mathbb{E}_{\mathbf{x} \sim \mathcal{D}} \left[ \mathbf{z}_t^\top \mathbf{z}_0 \right] = M \cdot tr(\mathbf{B}_0^\top \mathbf{B}_t) \tag{26}$$

$$\mathbb{E}_{\mathbf{x} \sim \mathcal{D}} \left[ \|\mathbf{z}_t\|_2^2 \right] = M \cdot tr(\mathbf{B}_t^\top \mathbf{B}_t) \tag{27}$$

### D.2 CRITICAL POINTS

Our goal is to find a good initialized task head, i.e., $\mathbf{v}_0$, which can lead to better downstream performance. Instead of directly linking $\mathbf{v}_0$ to the expected risk, which is a common practice for generalization analysis [5], we consider this problem from another indirect way. Specifically, we assume the SGD algorithm with appropriate regularization in FT stage can find a good optimum for the learning task. What we care more about is whether the features learned from the pretraining stage adapts well to the downstream task. We believe that if the features are properly adapted to the new environment, the model has more potential to generalize better. Hence in this part, considering the aforementioned three distance metrics as the target functions, we formalize how $\mathbf{v}_0$ influence them.

From Equation (25), (26) and (27), we find that the behavior of $tr(\mathbf{B}_0^\top \mathbf{B}_t)$ and $tr(\mathbf{B}_t^\top \mathbf{B}_t)$ are the keys. In these two terms, $\mathbf{B}_0$ is given and cannot change, while $\mathbf{B}_t$ is determined by the choice of $\mathbf{v}_0$. One way to link these two quantities is using Lemma A.4 in Kumar et al. (2022) or theorem 2.2 in Du et al. (2018):

$$\mathbf{v}_0 \mathbf{v}_0^\top - \mathbf{B}_0 \mathbf{B}_0^\top = \mathbf{v}_t \mathbf{v}_t^\top - \mathbf{B}_t \mathbf{B}_t^\top, \quad \forall t, \tag{28}$$

however, the expression of $\mathbf{v}_t$ is still hard to obtain (but collecting and visualizing $\mathbf{v}_t$ is much more cheaper than $\mathbf{B}_t$). We left this direction for our future work. In this paper, we analyze the problem in the NTK regime (this is the main assumption of our analysis, which can cause discrepancies between the theory and experiments).

**Closed-form of parameters under NTK approximation:**

To get more insights, we approximate the behavior of this model in the NTK regime, in which the converged parameters can be analytically calculated. Specifically, be applying Equation (8) in Lee et al. (2019) and assuming $t \to \infty$, we can have:

$$\theta_t = \theta_0 - (\nabla_\theta \mathbf{q}_0)^\top \mathcal{K}_0^{-1} (\mathbf{q}_0 - Y), \tag{29}$$

where $\mathbf{q}_0 \in \mathbb{R}^{N \times 1}$ is the model's prediction on $N$ training samples, i.e., $X$, and $\theta_t, \theta_0 \in \mathbb{R}^{(d+1)*h \times 1}$ are the stacked parameters. Without loss of generality, the first $d * h$ parameters in $\theta$ come from $\mathbf{B}$ and the last $h$ parameters come from $\mathbf{v}$.

The $\mathcal{K}_0 = \nabla_\theta \mathbf{q}_0 \cdot (\nabla_\theta \mathbf{q}_0)^\top \in \mathbb{R}^{N \times N}$ here is the empirical NTK on $X$. Specifically, by stacking the paramters, we can calculate each elements in this kernel as:

$$\kappa(\mathbf{x}, \mathbf{x}') = \mathbf{x}^\top \mathbf{B}^\top \mathbf{B} \mathbf{x}' + \sum_{i=1}^{h} \sum_{j=1}^{d} (v_i x_j)(v_i x_j')$$

$$= \mathbf{x}^\top \mathbf{B}^\top \mathbf{B} \mathbf{x}' + \sum_{i=1}^{h} v_i^2 \sum_{j=1}^{d} x_j x_j'$$

$$= \mathbf{x}^\top (\mathbf{B}^\top \mathbf{B} + \|\mathbf{v}\|_2^2 \cdot I_{d \times d}) \mathbf{x}', \tag{30}$$

where $v_i$ and $x_i$ is the $i$-th element in $\mathbf{v}$ and $\mathbf{x}$ respectively. Then, the matrix form of emperical NTK is $\mathcal{K}_0 = X(\mathbf{B}_0^\top \mathbf{B}_0 + \|\mathbf{v}_0\|_2^2 \cdot I_{d \times d}) X^\top$.

With the help of Equation (29), we can get the closed-form expression of $\mathbf{b}_t \triangleq \text{vec}(\mathbf{B}_t)$ (i.e., the vectorization of matrix $\mathbf{B}_t$):

$$\mathbf{b}_t = \mathbf{b}_0 - (\nabla_\mathbf{b} \mathbf{q}_0)^\top \mathcal{K}_0^{-1} (\mathbf{q}_0 - Y), \tag{31}$$

In Equation (31), we know $\mathbf{q}_0 = X \mathbf{B}_0^\top \mathbf{v}_0$. The term $\nabla_\mathbf{b} \mathbf{q}_0$ also depends on $\mathbf{v}_0$. As controlling $\mathbf{q}_0$, i.e., the model's prediction on training samples, is easier in practice (we can directly observe the training loss or training accuracy), we will consider $\mathbf{q}_0$ as the optimizing variable in the rest of the paper. The following lemma can link $\nabla_\mathbf{b} \mathbf{q}_0$ to $\mathbf{q}_0$:

---

[5]But it is hard to get tight and informative bounds in deep learning, especially in such a practical scenario.

**Lemma 1.** *For $\mathbf{b}_0 \in \mathbb{R}^{h*d \times 1}$ and $\nabla_{\mathbf{b}}\mathbf{q}_0 \in \mathbb{R}^{N \times h*d}$, we have $\nabla_{\mathbf{b}}\mathbf{q}_0 \cdot \mathbf{b}_0 = \mathbf{q}_0$.*

*Proof.* We check that equation elementwise. For the $n$-th element in the RHS, we have:

$$
\begin{aligned}
q_n &= \mathbf{x}_n^\top \mathbf{B}_0^\top \mathbf{v}_0 \\
&= \sum_{i=1}^{d} \sum_{j=1}^{h} x_i v_j (\mathbf{B}_0^\top)_{i,j} \\
&= \sum_{i=1}^{d} \sum_{j=1}^{h} x_i v_j (\mathbf{B}_0)_{j,i},
\end{aligned}
\tag{32}
$$

where $q_n$ is the $n$-th element of $\mathbf{q}_0$, $x_i$ is the $i$-th element of $\mathbf{x}_n$, and $v_j$ is the $j$-th element of $\mathbf{v}_0$.

For the LHS, the $n$-th value is $\mathbf{e}_n^\top \nabla_{\mathbf{b}}\mathbf{q}_0 \cdot \mathbf{b}_0$, where $\mathbf{e}_n^\top$ is a one-hot row vector selecting the $n$-th row of $\nabla_{\mathbf{b}}\mathbf{q}_0$. From the definition, we know $\mathbf{e}_n^\top \nabla_{\mathbf{b}}\mathbf{q}_0 = \mathrm{vec}(\mathbf{v}_0 \mathbf{x}_n^\top)^\top = [x_1 \mathbf{v}_0^\top, x_2 \mathbf{v}_0^\top, ..., x_d \mathbf{v}_0^\top]$, which is a long row vector. Then, the $n$-th element in the LHS should be:

$$
\begin{aligned}
\mathbf{e}_n^\top \nabla_{\mathbf{b}}\mathbf{q}_0 \cdot \mathbf{b}_0 &= \mathrm{vec}(\mathbf{v}_0 \mathbf{x}_n^\top)^\top \cdot \mathbf{b}_0 \\
&= \sum_{l}^{h*d} (\mathrm{vec}(\mathbf{v}_0 \mathbf{x}_n^\top))_l (\mathbf{b}_0)_l \\
&= \sum_{i=1}^{d} \sum_{j=1}^{h} x_i v_j (\mathbf{B}_0)_{j,i} \\
&= q_n,
\end{aligned}
\tag{33}
$$

where the last equation holds the rule of stacking elements in a matrix. As each elements of the two sides are equal, the lemma holds. $\qquad\square$

**Critical points of $tr(\mathbf{B}_0^\top \mathbf{B}_t)$ and $tr(\mathbf{B}_t^\top \mathbf{B}_t)$:**

**Lemma 2.** $\mathbf{q}_0^* = \frac{1}{2}Y$ *is a maximum of $tr(\mathbf{B}_0^\top \mathbf{B}_t)$.*

*Proof.* By definition, we have:

$$
\begin{aligned}
tr(\mathbf{B}_0^\top \mathbf{B}_t) &= \mathbf{b}_0^\top \mathbf{b}_t \\
&= \mathbf{b}_0^\top \left( \mathbf{b}_0 - (\nabla_{\mathbf{b}}\mathbf{q}_0)^\top \mathcal{K}_0^{-1} (\mathbf{q}_0 - Y) \right) \\
&= \mathbf{b}_0^\top \mathbf{b}_0 - (\nabla_{\mathbf{b}}\mathbf{q}_0 \cdot \mathbf{b}_0)^\top \mathcal{K}_0^{-1} (\mathbf{q}_0 - Y) \\
&= \mathrm{const.} - \mathbf{q}_0^\top C_1 (\mathbf{q}_0 - Y),
\end{aligned}
\tag{34}
$$

where we call $C_1 = \mathcal{K}_0^{-1}$ for convenience. By taking first derivative to $\mathbf{q}_0$ and let is equal zero, we know $\mathbf{q}_0^* = \frac{1}{2}Y$ is a critical point. As the second derivative is $-2C_1$ and $C_1$ is positive definite, this critical point is a maximum. $\qquad\square$

**Lemma 3.** $\mathbf{q}_0^* = \alpha Y, \alpha \in (0, 0.5)$ *is a critical point (usually maximum) of $tr(\mathbf{B}_t^\top \mathbf{B}_t)$. Here $\alpha = \left( I_{N \times N} + (C_2 - 2C_1)^{-1} C_1 \right) Y$, where $C_2 = \mathcal{K}_0^{-1} \tilde{\mathcal{K}}_0 \mathcal{K}_0^{-1}$ and $\tilde{\mathcal{K}}_0 = X \mathbf{B}_0^\top \mathbf{B}_0 X^T = (\nabla_{\mathbf{v}}\mathbf{q}_0)(\nabla_{\mathbf{v}}\mathbf{q}_0)^\top$ is the NTK when fixing the parameters of the backbone.*

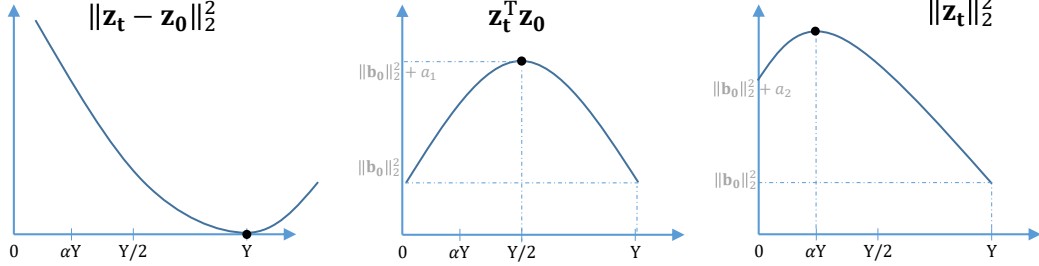

Figure 16: Illustrations of different metrics if they are 1-D quadratic functions. Here $a_1 = \alpha Y^\top C_1 Y$ and $a_2 = Y^\top C_2 Y$. The x-axis represents the choice of $\mathbf{q}_0$, where $\alpha Y$ is an example of the critical point of $\|\mathbf{z}_t\|_2^2$. The black dots are the critical points.

*Proof.* Similar to Lemma 2, we can write $tr(\mathbf{B}_t^\top \mathbf{B}_t)$ as $\mathbf{b}_t^\top \mathbf{b}_t$ and substitute Equation (31):

$$tr(\mathbf{B}_t^\top \mathbf{B}_t) = \mathbf{b}_t^\top \mathbf{b}_t$$
$$= \left(\mathbf{b}_0 - (\nabla_\mathbf{b}\mathbf{q}_0)^\top \mathcal{K}_0^{-1}(\mathbf{q}_0 - Y)\right)^\top \left(\mathbf{b}_0 - (\nabla_\mathbf{b}\mathbf{q}_0)^\top \mathcal{K}_0^{-1}(\mathbf{q}_0 - Y)\right)$$
$$= \mathbf{b}_0^\top \mathbf{b}_0 - 2(\nabla_\mathbf{b}\mathbf{q}_0 \cdot \mathbf{b}_0)^\top \mathcal{K}_0^{-1}(\mathbf{q}_0 - Y) + (\mathbf{q}_0 - Y)^\top \mathcal{K}_0^{-1}(\nabla_\mathbf{b}\mathbf{q}_0)(\nabla_\mathbf{b}\mathbf{q}_0)^\top \mathcal{K}_0^{-1}(\mathbf{q}_0 - Y)$$
$$= \mathbf{b}_0^\top \mathbf{b}_0 - 2\mathbf{q}_0^\top \mathcal{K}_0^{-1}(\mathbf{q}_0 - Y) + (\mathbf{q}_0 - Y)^\top \mathcal{K}_0^{-1}\tilde{\mathcal{K}}_0\mathcal{K}_0^{-1}(\mathbf{q}_0 - Y)$$
$$= \mathbf{b}_0^\top \mathbf{b}_0 - 2\mathbf{q}_0^\top C_1(\mathbf{q}_0 - Y) + (\mathbf{q}_0 - Y)^\top C_2(\mathbf{q}_0 - Y)$$
$$= (\mathbf{b}_0^\top \mathbf{b}_0 + Y^\top C_2 Y) + \mathbf{q}_0^\top(2C_1 - 2C_2)Y + \mathbf{q}_0^\top(C_2 - 2C_1)\mathbf{q}_0$$
$$= \text{const.} + \mathbf{q}_0^\top(2C_1 - 2C_2)Y + \mathbf{q}_0^\top(C_2 - 2C_1)\mathbf{q}_0 \tag{35}$$

By taking first derivative and letting it equal zero, assuming $(C_2 - 2C_1)$ is invertible, we know $\mathbf{q}_0^* = \left(I_{N \times N} + (C_2 - 2C_1)^{-1}C_1\right)Y$ is a critical point. The second derivative is $2C_2 - 4C_1$, which is usually positive definite in our settings (explain later). $\qquad \square$

To get more insights, we can look deeper into the critical point mentioned in Lemma 3. Following the definition of $C_1$ and $C_2$, we can have:

$$C_2 - 2C_1 = \mathcal{K}_0^{-1}\tilde{\mathcal{K}}_0\mathcal{K}_0^{-1} - 2\mathcal{K}_0^{-1}$$
$$= (\mathcal{K}_0^{-1}\tilde{\mathcal{K}}_0 - 2I_{N \times N})\mathcal{K}_0^{-1}, \tag{36}$$

where $\mathcal{K}_0^{-1}\tilde{\mathcal{K}}_0$ is the key of understanding this term.

The exact form of this expression is hard to obtain, but as all the NTK or covariance matrices mentioned here are sysmetric, we can compare the trace of them to get some insights. By definition, $tr(\tilde{\mathcal{K}}_0) = tr(X\mathbf{B}_0^\top \mathbf{B}_0 X^\top) = tr(\mathbf{B}_0^\top \mathbf{B}_0 X^\top X)$. If each dimensions of the data samples are independent, then $X^\top X \approx I_{d \times d}$, and $tr(\tilde{\mathcal{K}}_0) \approx tr(\mathbf{b}_0^\top \mathbf{b}_0) = \|\mathbf{b}_0\|_2^2$. Then similarly, $tr(\mathcal{K}_0) \approx \|\mathbf{b}_0\|_2^2 + \|\mathbf{v}_0\|_2^2$. Thus the behavior of term $\mathcal{K}_0^{-1}\tilde{\mathcal{K}}_0$ can be described by $\frac{\|\mathbf{b}_0\|_2^2}{\|\mathbf{b}_0\|_2^2 + \|\mathbf{v}_0\|_2^2} \cdot I_{N \times N}$:

- if $\|\mathbf{b}_0\|_2^2 \gg \|\mathbf{v}_0\|_2^2$, which might happen as the backbone contains more parameters than the head, $\mathcal{K}_0^{-1}\tilde{\mathcal{K}}_0 \approx I_{N \times N}$,
- if $\|\mathbf{b}_0\|_2^2 \ll \|\mathbf{v}_0\|_2^2$, which might happen if we split the network in the earlier layer, $\mathcal{K}_0^{-1}\tilde{\mathcal{K}}_0 \approx 0 \cdot I_{N \times N}$.

But in either case, the negative of the second derivative of $tr(\mathbf{B}_t^\top \mathbf{B}_t)$, i.e., Equation (36), is likely to be positive definite, which means the critical point in Lemma 3 is usually a maximum. Another interesting fact of Lemma 3 is $\mathbf{q}_0^*$ under the aforementioned two extreme conditions. After some calculation, we can verify that $\mathbf{q}_0^* \to 0 \cdot Y$ if $\|\mathbf{b}_0\|_2^2 \gg \|\mathbf{v}_0\|_2^2$ and $\mathbf{q}_0^* \to \frac{1}{2}Y$ if $\|\mathbf{b}_0\|_2^2 \ll \|\mathbf{v}_0\|_2^2$.

**Critical points of $\bar{d}_{euc}$, $\bar{d}_{dot}$ and $\bar{d}_{cos}$:**

Recall the definitions of different distance metrics we care about. First, $\bar{d}_{dot}$ is proportional to $\mathbf{z}_t^\top \mathbf{z}_0$, hence its shape would like the second panel in Figure 16. For $\bar{d}_{euc}$, we know it is proportional to $tr(\mathbf{B}_t^\top \mathbf{B}_t) - 2tr(\mathbf{B}_0^\top \mathbf{B}_t) + tr(\mathbf{B}_0^\top \mathbf{B}_0)$. Substituting results in Equation (34) and Equation (35), we have:

$$
\begin{aligned}
\bar{d}_{euc} &\propto tr(\mathbf{B}_t^\top \mathbf{B}_t) - 2tr(\mathbf{B}_0^\top \mathbf{B}_t) + \text{const.} \\
&= \mathbf{q}_0^\top (2C_1 - 2C_2)Y + \mathbf{q}_0^\top (C_2 - 2C_1)\mathbf{q}_0 + 2\mathbf{q}_0^\top C_1(\mathbf{q}_0 - Y) + \text{const.} \\
&= -2\mathbf{q}_0^\top C_2 Y + \mathbf{q}_0^\top C_2 \mathbf{q}_0 + \text{const.}
\end{aligned}
\tag{37}
$$

Obviously, $\mathbf{q}_0^* = Y$ is the minimum of $\bar{d}_{euc}$, as depicted in the first panel in Figure 16. The shape of $\bar{d}_{cos}$ is hard to obtain. However in the last panel of Figure 16, we demonstrate $\|\mathbf{z}_t\|_2^2$, its denominator, to assist our further analysis.

### D.3 LEARNING DYNAMICS OF THE REPRESENTATION

| $\mathbf{q}_0 =$ | $E_{aie}$ | $d_{euc}$ | $d_{dot}$ | $\|\mathbf{z}_t\|_2$ | $d_{cos}$ |
|---|---|---|---|---|---|
| $Y \to \frac{1}{2}Y$ | ↑ | ↑ | ↑ | ↑ | ? (↓) |
| $\frac{1}{2}Y \to \alpha Y$ | ↑ | ↑ | ↓ | ↑ | ↓↓ |
| $\alpha Y \to 0$ | ↑ | ↑ | ↓ | ↓ | ? (↓) |

Table 9: Question marks in the last column means we cannot accurately predict its change. But it usually decreases in experiments.

In this subsection, we will put everything together to provide an overview of how $\mathbf{z}_t$ changes compared with $\mathbf{z}_0$. Remember when $\mathbf{q}_0 = Y$, the gradient of $\mathbf{B}_0$ is zero, hence $\mathbf{z}_t = \mathbf{z}_0$ (Kumar et al., 2022). When $\mathbf{q}_0$ moves linearly from $Y$ to 0, we have the following three phases (see Table 9):

- $Y \to \frac{1}{2}Y$: $\mathbf{z}_t$ lengthen its norm with a slightly change in the direction, hence $\bar{d}_{dot}$ also increase;

- $\frac{1}{2}Y \to \alpha Y$: the norm of $\mathbf{z}_t$ keeps increasing, but its direction drastically changes in this phase, which makes $\bar{d}_{dot}$ decrease;

- $\alpha Y \to 0$: the norm of $\mathbf{z}_t$ begins to decrease and the angle between $\mathbf{z}_t$ and $\mathbf{z}_0$ keeps increasing, which makes $\mathbf{z}_t$ changes a lot.

