# OpenReview forum: "How to prepare your task head for finetuning"
_ICLR.cc/2023/Conference — ICLR 2023 poster_

### Official Review · Reviewer_5UFU · 2022-10-25

**Confidence:** 4
**Correctness:** 4
**Technical Novelty And Significance:** 3
**Empirical Novelty And Significance:** 3
**Recommendation:** 8

**Clarity, Quality, Novelty And Reproducibility:**

clarity:  although the subject matter is quite basic, some of the elements involved require in-depth theoretical knowledge, e.g. theory
behind the empirical neural tangent kernel (NTK). Experiments should have been better organized.
novelty: the approach is novel
originality: is original

**Strength And Weaknesses:**

strenghts:

The paper is well written and the problem is clearly stated multiple times
● Every consideration is presented both in a discussion and in a formal fashion
● Several comparisons are made with other similar analyses, outlining the differences in the proposed approach
● When realized (see the weaknesses), the experiments are extensive and well commented
● The topic of the paper is very relevant, and such an analysis, if done comprehensively, can be very useful, theoretically and practically, to many researchers in building better models and learning procedures

weaknesses

Sometimes the claims and observations of the authors seem a bit too excessive in comparison with the results actually obtained from the experiments. Some example are:
○ “Although we do not analyze the relationship between the change of zt and out-of-distribution performance – this adds too many “moving parts” to the current setting – our analysis is also relevant to finding features which will work in new, unseen domains” in Section 4. I think that out of distribution analysis is necessary to get a more thorough idea about the interaction between HP and FT, as stated in the highly mentioned Kumar et al., 2022
○ The paper lacks experiments regarding the relationship between backbone depth and head capacity. There are some examples in the appendix but they are too limited, considering that the use of non-linear and more complex heads (instead of the more basic linear probing) is one of the 3 practical suggestions given in the introduction
● The experimental section lacks some information (that can often be found in the appendix, but that I consider essential for the fruition of the paper):
○ A description, even short, of the used architectures (and how they are decomposed in backbone and head)
○ A small description of the considered configurations, for example of the Byol and MoCo unsupervised cases
○ In table 1 it is not clear to me what HP𝛕*-FT means. If it means the configuration with the best 𝛕, how 𝛕 was chosen?
● Some images (Figure 8 and 11) in the appendix are cited in the paper. Since these are quite useful perhaps it would be appropriate to include them somehow in the main paper
● I think the difference between the dataset used for pre-training and the dataset used for fine-tuning is a key element of such an analysis. This and what is meant by distance between tasks are elements that should be explored further in the main paper in my opinion.

**Summary Of The Paper:**

As clearly stated in the title, the paper presents an extensive analysis of how to prepare (or train) a task head in a backbone + task head neural network configuration. The authors also make a lot of reference to the cited paper by Kumar et al., frequently pointing out the differences in their approach (2022). The authors analyze the behavior of features after model pre-training and during the head probing (HP) phase, in which only the parameters of the head are updated, and the fine-tuning (FT) of the entire network. The analysis is carried out both with a toy example and also with more general downstream real tasks. By decomposing the learning dynamics of the representation
extracted by the backbone, they find the keys are the energy and direction terms, which highly correlate with the accuracy at the beginning of the finetuning process, then they show how to manipulate the energy (e.g., stop the Head Probing stage earlier, label smoothing only in Head Probing, non-linear head, etc.) to enhance the downstream performance.

**Summary Of The Review:**

The paper could have been definitely better with a stronger experimental section (see my comments above.)

POST REBUTTAL:
The paper is now satisfying, the User Guide section is useful, many explanations have been provided.

---

> ### Author Response · Authors · 2022-11-13
> **Change the paper a lot, not sure whether OOD is so important for this version 1/1**
>
> Thank you for identifying the strengths of this work. We appreciate the thorough review and have made several enhancements based on your comments. If you have other concerns in mind, we would be happy to take them into consideration.
>
> Please also note the [**overall response**](https://openreview.net/forum?id=gVOXZproe-e&noteId=gSQsN_eyVRJ). Here is the piece-to-piece response.
>
> #### Weakness:
>
> > Sometimes the claims and observations of the authors seem a bit too excessive …
>
> Thanks very much for the useful comment. We delete such excessive conclusions in the current version and only focus on in-distribution generalization. We agree that analyzing OOD can strengthen the claims of the paper, but clearly defining OOD is also an open problem (we already have too many open problems here, like the distance between PT and DS, how much we need to adapt the features, etc.). So we are a bit reluctant to add them in the current version. Actually, we already have some results of pretraining on IN1K and transferring to Domain-Real. Then should we consider Domain-Sketch/Quick as the OOD tasks or some completely different tasks? Furthermore, for the OOD problem, we only add a test set for the out-distribution. As the focus of the paper is on the interaction between task head and backbone while fine-tuning, the OOD test set won’t influence the learning process as well. In summary, we can add some discussions and experiments in the appendix if necessary, but we are afraid the main contribution of the paper would not be changed so much.
>
> > The paper lacks experiments regarding the relationship between backbone depth …
>
> Thanks very much for your suggestions, we now discuss these two tricks in section 5.3. For the label smoothing, we elaborate on our motivation and add an interesting ablation experiment in table2. For the MLP head, we keep it in table 2 and add some discussions in the appendix. For the backbone depth, we put table 3 in the main text. Some similar results are in the appendix. The experiments for these tricks are a little limited as we find it only works on a few specific settings (so we suggest using this trick only when you have strong belief), like backbone is not well pretrained (the CIFAR100 pretraining case) or the downstream tasks require too simple features (like Domain-Quick). So we believe the current experiments are enough to show when to consider these tricks and how these tricks influence energy. Anyway, we can add more supporting experiments in the next version if the reviewer thinks those results are really helpful.
>
> > The experimental section lacks some information … A description, even short, of the used architectures …
>
> Thanks very much for the suggestion. We add that in Appendix B in this version.
>
> > In table 1 it is not clear to me what HP𝛕*-FT means …
>
> Thanks for the comment. We fix this in the current version. It means first HP $\tau$ epoch and then FT to converge. $\tau$ is selected using a validation split. All the accuracy numbers are test accuracy in this version.
>
> > Some images (Figure 8 and 11) in the appendix are cited in the paper …
>
> Thanks for the suggestion. Now the flow of the paper is changed, so we don't cite them in the main text anymore.
>
> > I think the difference between the dataset used for pre-training and the dataset used for fine-tuning is a key element of such an analysis …
>
> Yeah, we totally agree with the reviewer on this point. The distance between PT and DS tasks is the origin of how much adaptation we need to perform well on downstream tasks. But defining this is really an open problem in the machine learning community, as the network might interpret the input in a totally different way than we humans do. Thus we rearrange the paper to bypass such a heuristic problem and highlight the takeaway message in Section 5 and Q1 in the overall response. So we suggest a phenomenon → hypothesis → solution workflow and use the validation performance to verify our belief. For example, if we believe the distance between PT and DS is large, we can then see whether strong adaptation is better than mild adaptation (or other advanced tricks). If the result is true, we can further increase the energy to see what will happen. If the results are opposite, that means the ‘actual distance’ is not as large as we imagine. Then we should try a smaller adaptation. Facing this complicated problem, this paper provides extra hyper-parameters and tricks to try. The theoretical analysis also ensures the generality of our suggestions. We also believe this work has the potential to inspire future works related to understanding the learning dynamics, features’ interoperability, etc.

---

### Official Review · Reviewer_6K2Z · 2022-10-25

**Confidence:** 3
**Correctness:** 3
**Technical Novelty And Significance:** 3
**Empirical Novelty And Significance:** 3
**Recommendation:** 6

**Clarity, Quality, Novelty And Reproducibility:**

The observations about the effect of number of HT steps on FT is novel and relevant, to the best of my knowledge. There are many issues regarding clarity that are described above.

**Strength And Weaknesses:**

**Strengths**

- Overall the paper presents an interesting analysis of the role of task head in the fine-tuning performance. The following observations are interesting: the optimal stopping criterion is somewhere in the middle and that label-smoothing during head-tuning can help get closer to the performance of this optimal stopping criterion.

- The paper also performs deeper analysis into some of these phenomena, e.g. analyzing the level of feature adaptation as a function of number of initial HT steps $\tau$ or "energy". Many of the hypotheses are verified empirically even if not theoretically proved.


**Weaknesses**

- A lot of the analysis is hand wavy. For e.g. the effect of the task head on the FT features is studied by just looking at just 1 step of gradient descent. The notion of "energy" derived from this is also vague since it is technically a vector. Although empirically heuristics like the training accuracy or probability gap seem to be a good proxy for "energy". The theoretical result for the linear model uses $\alpha$ in $\alpha Y$ as a proxy for "energy". Despite all of these, the analysis seems somewhat meaningful.

- I think the presentation of the paper can be significantly improved. I found the paper hard to follow and it took me at least 2 passes to understand how the different points in the paper connect to each other. There is a constant switching from number of HT steps being important to the "energy" being crucial to feature adaptation being the key. Rather than presenting the analysis first, I think it might help to start with the empirical observation that the ideal choice of task head is to initially HT for some $\tau^*$ number of steps before FT. Then the goal of the entire study could be to understand this phenomenon and try to achieve this optimal performance in a simple way (e.g. label-smoothing is a solution). Without such a concrete goal, I was lost many times about the purpose of different sections.

Other comments/questions:
- If my interpretation is correct, it is a bit misleading to suggest in Proposition 1 (informal) that the result is about the effect of $\tau$, because the X-axis seems to be something different, $\alpha$ in $\alpha Y$
- "This kind of z can make the original overlapped features become separable after adapting to a similar manifold" on page 5. Could you please elaborate what this means?
- How was the label-smoothing coefficient of 0.9 picked? How much of an effect does this have?
- The discussion in Section 3.3 about head capacity seems to be hand-wavy and a distraction, since most of the results seem to use a linear head

**Summary Of The Paper:**

This paper studies the role of choice of task head, i.e. head tuning (HT), in the downstream performance of fine-tuning (FT) from a pre-trained model.
Recent work has suggested that it is a good idea to first perform head tune on a downstream task and then use that head for fine-tuning (so that features do not move too much), rather than fine-tuning from a randomly initialized head (where features move a lot).
This paper also considers an intermediate case of stopping early during head tuning (to mildly adapt features) and finds empirically that in many cases the optimal stopping criterion is somewhere in the middle.

The paper delves deeper into the amount of feature adaptation w.r.t. number of steps $\tau$ of HT performed initially, and find some interesting trends like norm of change in features $\|z\_t - z\_0\|$ is monotonically decreasing w.r.t. $\tau$ while the norm of the features $\|z\_t\|$ has a quadratic behavior. Theory on an overparametrized linear setting (NTK regime) that tries to explain these trends.
The analysis of 1 step of feature dynamics highlights an “energy” term that strongly determines the amount of feature adaptation: energy is low when initial HT on top of pre-trained features already has good accuracy and vice versa.


Using these findings and analysis, the paper identifies label smoothing during HT as a way to control the “energy” and allow feature adaptation. This often (but not always) performs comparably in practice to the optimal $\tau$ (number of HT steps) which is not easy to find.

**Summary Of The Review:**

Overall I find the contributions of this paper in understanding the role of task head in fine-tuning to be positive. However due to the presentation concerns I have only assigned a score of weak accept.

---

> ### Author Response · Authors · 2022-11-13
> **Further justify most of the claims 1/1**
>
> Thank you for identifying the strengths of this work. We appreciate the thorough review and have made several enhancements based on your comments. If you have other concerns in mind, we would be happy to take them into consideration.
>
> Please also note the [**overall response**](https://openreview.net/forum?id=gVOXZproe-e&noteId=gSQsN_eyVRJ). Here is the piece-to-piece response.
>
> ### Weakness:
>
> > A lot of the analysis is hand-wavy. For e.g. the effect of the task head on the FT features is studied by just looking at just 1 step of gradient descent. …
>
> We thank the reviewer point out this important question. In this version, we go further from $z_{t+1}-z_t$ to $|z_T-z_0|$ and use Average Initial Energy to bound that (see proposition 1). This also provides a formal definition of energy. We use this AIE as the pivot to connect HP-train-acc, $\tau$, and $\alpha$ in the toy example. We can consider these hyper-parameters or tricks influence $p_0$, hence influencing AIE, which in turn influences $z$’s adaptation. HP-train-acc can be considered an observation of AIE.
>
> > I think the presentation of the paper can be significantly improved. …
>
> Thanks very much for the useful comment. The flow of the paper is changed following your suggestions. Please refer to Q0 in the overall response.
>
> > If my interpretation is correct, it is a bit misleading to suggest in Proposition 1…
>
> That’s a good point. The current version first proposes the concept of AIE, then links all other quantities to that. After claiming $\tau$ positively correlated with HP-train-acc, which positively correlated with AIE, we can use $\tau$ as the x-axis. The value of $\alpha$ also links to AIE in this way (see the last table in the appendix).
>
> > This kind of z can make the original overlapped features become separable after adapting …
>
> Thanks for the comment. Please refer to Q2 in the overall response and section 4.2 in the new version.
>
> > How was the label-smoothing coefficient of 0.9 picked? …
>
> Yeah, we just pick a not-too-bad value for the experiments. We can definitely tune this hyper-parameter for better results, but it helps little with the story of this paper. The current version discusses more the label smoothing part, please refer to section 5.3.
>
> > The discussion in Section 3.3 about head capacity seems to be hand-wavy…
>
> Thanks for pointing out this. Those discussions (now section 4.3) are indeed quite high-level and based on intuitions. So we treat them as the last tricks in our paper. In the current version, we add more experiments in the main text (Tables 2, and 3) to justify it. (The head of a segmentation task is also quite complex). However, as the partial backbone and heavy-head settings involve too many variables, it is really hard for us to make further conclusions. Anyway, we see those results still match our basic claims quite well.

---

> > ### Comment · Reviewer_6K2Z · 2022-12-09
> > **Response to authors**
> >
> > Thank you for the response and the revision. The new re-structured version, with the user guide in the introduction and motivation in Section 2, will definitely improve the flow of arguments. The new Proposition 1 (which didn't exist earlier) is also a good addition. I have the following follow-up comments on Proposition 2 (which used to be Proposition 1).
> >
> > I still believe that Proposition 2 (informal), even with the new definition for energy, is a bit misleading in how it is phrased. Additionally there is no "formal" Proposition 2 anywhere in the paper. As far as I understand the result, there is no a direct connection between $\tau$ (what is the intended X-axis) and $\alpha$ (what is proved in the result). In fact there is no $\tau$ (number of HP steps) in the NTK analysis anywhere. There is only a vague connection because both $\tau$ and $\alpha$ can be positively correlated with AIE, but it is not clear why a result for $\alpha$ also extends to $\tau$, at least in any formal sense. Either the connection between $\tau$ and $\alpha$ needs to be made more explicit, or the statement of Proposition 2 and the discussion around it should provide some explanation. It would be useful to have some sort of formal statement describing the exact setting in which the analysis was performed, and the precise results. Otherwise it is hard to judge if the claim "We analytically prove this trend" from the abstract is well justified.
> >
> > On re-reading the analysis more closely, I'm not sure I followed Eq 25. Is $E[xx^T]$ assumed to be $M .\text{Id}$ and why? If $M$ is a matrix then it cannot be taken out of the trace operation. Again a formal statement would help.
> >
> > Overall, given the volume of changes (in structure, content, definitions, results) made in the revision, it was a little difficult to keep track of all them and appropriately re-evaluate the paper. Thus I am hesitant to increase my score beyond weak accept.

---

> > > ### Author Response · Authors · 2022-12-09
> > > **Thanks for further question.**
> > >
> > > Thanks very much for your feedback and further question. We agree that the correlation between $\alpha$, $\tau$, and AIE is a bit confusing in the current version. In our mind, the causal relationship among them might be $\tau\rightarrow\text{train-acc}\rightarrow$ AIE $\leftarrow\alpha$, i.e., either $\alpha$ and $\tau$ can cause the change of AIE. Hence we can claim that $\tau$ and $\alpha$ are correlated.
> > >
> > > Another big gap between our analysis of the toy case and the real case is that we are analyzing a regression (or binary classification) in the toy case but conducting multi-class classification in the real case. That is why we have $\alpha\cdot Y$ in the analysis, which is hard to relate to the training accuracy. We plan to analyze a toy multi-class classification problem and see whether things can be better.
> > >
> > > For equation 25, thanks for pointing this mistake out. One thing we can remedy this is changing the equation to an upper bound, from the fourth line:
> > >
> > > $ = tr(x^T B_t^T B_t x-2x^T B_0^T B_t x + x^T B_0^T B_0 x)$
> > >
> > > $ = tr(x^T (B_t^T B_t - 2 B_0^T B_t +  B_0^T B_0) x)$
> > >
> > > $ = tr((B_t^T B_t - 2 B_0^T B_t +  B_0^T B_0) x x^T )$
> > >
> > > $ = tr((B_t^T B_t - 2 B_0^T B_t +  B_0^T B_0) M )$
> > >
> > > $ \leq tr(B_t^T B_t - 2 B_0^T B_t +  B_0^T B_0) \cdot ||M||_{op}$
> > >
> > > or
> > >
> > > $ \leq ||B_t^T B_t - 2 B_0^T B_t +  B_0^T B_0)||_F \cdot ||M||_F$,
> > >
> > > using two versions of Hölder's inequality for the trace at the end.
> > >
> > > Then all the analysis still holds.

---

### Official Review · Reviewer_kiZg · 2022-11-01

**Confidence:** 4
**Correctness:** 3
**Technical Novelty And Significance:** 2
**Empirical Novelty And Significance:** 2
**Recommendation:** 5

**Clarity, Quality, Novelty And Reproducibility:**

The paper is well written. The novelty of the paper lies at understanding the transfer learning performance by analyzing two key terms with different conditions.

**Strength And Weaknesses:**

Strength
- The paper is well structured and written.
- The paper characterizes the change of the backbone and head during finetuning, which provides a better understanding about the evolving process of feature adaptation during fine-tuning.

Weakness
- The authors made observations on the quadratic trend of $z_t^Tz_0$ and $\|z_0\|_2^2$, however, it is not clear how this trend can help adapt features on the downstream task.
- The authors claimed that the change of the NTK term is slow during FT without further examination. Is it possible that the NTK term also matters when the source and target data are different and the change could be large?
- How much to adapt may also depends on the dataset size, which were not considered.
- There are no recommendations for how to select the backbone depth or manipulate the head’s capacity. The selection of $\tau$ is also not clear.


**Summary Of The Paper:**

This paper studies how the choice of task heads influence the pretrained features z’s adaptation and hence influence the downstream performance.  The authors explain feature adaptation by decomposing the learning dynamics and identify the energy and direction terms matter most. They find that the training accuracy and loss at the beginning of finetuning determines the energy for the feature’s adaptation. They also identify a trend in the effect of changes in this initial energy on the resulting features after fine-tuning, which helps to design an appropriate fine-tuning procedure. Finally they provide examples of how to enhance downstream performance in practice. They suggest paying more attention to how much and in what ways we want the features to adapt.


**Summary Of The Review:**

The paper presents an analysis of the fine-tuning behavior with head probing in different settings. However, it is not clear whether the analysis can lead to a concrete approach or recommendation for controlling feature adaption.

---

> ### Author Response · Authors · 2022-11-13
> **Change the paper a lot, not sure whether lazy NTK and dataset influence are so important 1/1**
>
> Thank you for identifying the strengths of this work. We appreciate the thorough review and have made several enhancements based on your comments. If you have other concerns in mind, we would be happy to take them into consideration.
>
> Please also note the [**overall response**](https://openreview.net/forum?id=gVOXZproe-e&noteId=gSQsN_eyVRJ). Here is the piece-to-piece response.
>
> #### Weaknesses:
>
> > The authors made observations on the quadratic trend of dot(zt,z0) and norm(zt),…
>
> Thanks for this question. We think the current section 4.2 and Q2 in the overall response can answer this question well. Specifically, by analyzing $\|z_T-z_0\|$, we can only conclude that larger energy leads to more adaptation. But the analysis of the other three quantities gives us a non-trivial trend of features’ adaptation, which helps us understand what happens during fine-tuning. Inspired by this trend, we provide lsHP as an alternative method to increase energy, because this method almost guarantees a mild adaptation (remember the strong adaptation is mainly caused by the noisy random direction phase when train-acc is low).
>
> > The authors claimed that the change of the NTK term is slow during FT …
>
> Thanks for pointing this out. Actually, the slow change NTK assumption is not that important in our analysis. In this version, the upper bound of $\|z_T-z_0\|$ only assumes the norm of NTK is bounded, which is usually true for a well-trained model. For the over-parameterized linear model, the kernel is just $X^T X \cdot I$, which is invariant during FT. For the real tasks finetuning, such a claim usually holds, because most of the information is already learned during pre-training. Furthermore, the learning rate is usually quite small in fine-tuning. As claimed in [1], when the learning rate is small, the kernels indeed change quite slowly. We are working on an experiment to show that compared with pre-training, the change of NTK is rather slow and small during fine-tuning, even in the strong adaptation case. If the reviewer believes this is really important to the current submission, we will add this in the next version.
>
> > How much to adapt may also depends on the dataset size, …
>
> Thanks very much for pointing this out. We totally agree with the point that how much to adapt may also depend on the downstream dataset size. However, we believe this is also a heuristic problem, similar to the distance between PT and DS datasets. Indeed, the influence of dataset size might be much more complex: it also depends on how similar the PT and DS datasets are and how capable the task head is. For the small dataset size specifically, if the task head is huge, we should not HP too much as the head might overfit the DS dataset. If the PT and DS datasets are quite similar, we should HP enough to ensure small energy. As this quantity is quite hard to analyze, we decided to delete that from the current version (actually in the original version before submission, we have a section discussing this). In this version, we bypass those hard-to-define problems (e.g., PT/DS difference, dataset size) and only provide some high-level discussions on that. The analysis cannot give a crystal clear guide on what to do facing different tasks, but it gives us useful suggestions when specific phenomena are observed, including the influence of the dataset size. If you indeed think those discussions are helpful for the current version, we can add some results by manipulating the ratio of the training samples in the finetuning in the next version.
>
> > There are no recommendations for how to select the backbone depth or manipulate the head’s capacity. The selection of tau is also not clear. …
>
> Thanks very much for the comments. The discussion of backbone depth, head’s capacity, and $\tau$’s selection are all elaborated in section 5 and Q1 in the overall response. We suggest first guessing what kind of adaptation is helpful when facing a task, then trying the basic method or advanced tricks. The best $\tau^*$ in our experiment is selected by sweeping and tuning on a validation split (first split a validation set from the training set, after choosing $\tau$, merge it back to the training set and train again).
>
> [1] Yang, Greg, and Edward J. Hu. "Feature learning in infinite-width neural networks." arXiv preprint arXiv:2011.14522 (2020).

---

> > ### Comment · Reviewer_kiZg · 2022-12-10
> > **Thanks for the response**
> >
> > I appreciate the authors' effort for answering my questions and revising the draft. However, some of my concerns for the work are not quite addressed.
> >
> > - non-trivial trend and gap estimation: it is not clear how the "non-trivial trend" helps to design the proposed of \tau sweeping strategy. Since the energy term is hard to predict for a new tasks, the \tau sweeping reduces to a normal HPO and can be developed without the analysis. The authors acknowledged that “this non-trivial trend might only slightly influence the final performance" and "believe our analysis and findings could be important for the works focusing on explainability or interpreting the features".
> >
> > - the NTK change: the comparison of NTK change for random initialized model and pre-trained model just strengthens my concern. For strong adaptation, as defined in the paper "pre-trained features are far from the optimal ones for downstream tasks and substantial feature adaptation is needed", the NTK norm indeed changed significantly even at the end of training after full convergence. It is not clear what dataset and pre-trained model were used for this comparison, it is possible in real fine-tuning cases  the target task can be irrelevant with pre-trained model (e.g., pre-trained on CIFAR10) or the target task is hard to solve (training acc will not reach zero), the pre-trained weights may act similar as random initialization and therefore the NTK terms could change quickly.
> >
> > - how to select \tau and other tricks: the paper is essentially proposing to perform sweeping for \tau and iterative trying of different strategies. However, the authors use different learning rates for HP for different tasks. It is not explained how optimal learning rate is selected for different tasks. It might indicate the necessity to search both learning rate and \tau for optimal performance, or one may fix \tau but perform lr sweeping. Actually the same sweeping strategy can be applied to other hyperparameters such as backbone depth and head’s capacity. I am not quite convinced given the simplicity of the datasets like flowers, STL10 and CIFAR10. I would like to see more thorough evaluation on diverse downstream tasks with different dataset sizes.
> >
> > Given that volume of changes in the vision and above reasons, I would keep my score but not against its acceptance.

---

> ### Author Response · Authors · 2022-11-17
> **Add a small experiment in Appendix A.**
>
> Hi, we add a small experiment in Appendix A in the new version to justify the assumption that the NTK term changes slowly during fine-tuning. Such an assumption is more likely to hold when we copy the parameters from a pretrained model (i.e., our fine-tuning case). We hope that will be helpful.

---

### Official Review · Reviewer_xD14 · 2022-11-02

**Confidence:** 2
**Correctness:** 4
**Technical Novelty And Significance:** 3
**Empirical Novelty And Significance:** 3
**Recommendation:** 6

**Clarity, Quality, Novelty And Reproducibility:**

Novelty seems mostly on the theoretical area and significant. The take away messages from some sections seem to be aligned with the empirical result, such as a higher energy needs a longer feature adaptation. Was there some take away message that go beyond the general empirical impression? Please let me know if I do not understand well

Clarity is good but maybe challenge to follow if the reader does not work on DL theory.

**Strength And Weaknesses:**

Strength
1) Extensive study for numerical experiments.
2) The energy decomposition seems novel.
3) The problem of how to make pretraining / fine tuning work better is very core, and thus the paper may seem of great interest to people in the area.

Question
The overall writeup seems challenge for a regular non-DL-theory practitioner to follow, even with the Section 4 on real world problems. Can you please make the implication / take away message a little more explicit for model training of real-world problem?

**Summary Of The Paper:**

The authors focus on the choice of task head in fine tuning, and how the task head controls feature adaptation and downstream model quality.

The gradient was decomposed as the product between a direction and an energy, and analysis was done on the effect of the energy vs feature adaptation after FT step.

The paper's extensive experiments would have guidance to how to use a few techniques to get better model, such as early stopping at HP, and label smoothing at HP.

**Summary Of The Review:**

Recommend as accept but marginally above threshold. If the authors could help me to appreciate better the paper's empirical value, then it would be awesome

---

> ### Author Response · Authors · 2022-11-13
> **Change the flow, add more experiments 1/1**
>
> Thank you for identifying the strengths of this work. We appreciate the thorough review and have made several enhancements based on your comments. If you have other concerns in mind, we would be happy to take them into consideration.
>
> Please also note the [**overall response**](https://openreview.net/forum?id=gVOXZproe-e&noteId=gSQsN_eyVRJ). Here is the piece-to-piece response.
>
> > Novelty seems mostly on the theoretical area and significant. The takeaway messages from …
>
> We thank the reviewer for the useful comments. IIUC, the reviewer has the following two concerns: what is the empirical takeaway message, and is there any non-trivial information that can support the analysis? (Please refer to Q1 and Q2 in the overall response).
>
> For the first point, we restructure the paper and use section 5.1 to highlight our practical suggestions. In short, we can consider the paper provides some useful tools for different practical situations. But when and how to use them depends on the specific tasks we are facing. We can compare the application of dropout or weight decay in solving overfitting issues. Following the “phenomena —> hypothesis —>solution pattern”, we know:
>
> ”Val-acc begins to decrease after converge —> model is overfitted —> try dropout”
>
> Similarly, for the feature’s adaptation, we have:
>
> ”train acc 100% after HP —> insufficient energy —> try esHP or lsHP”.
>
> Although the real conditions can be more complex, we hope the user guide provided can help the practitioners to understand when and why to apply these methods. Additionally, we also add experiments on image segmentation to support our analysis and enlarge the scope of our methods.
>
> For the second point, we also rewrite section 4.2 to highlight this. One thing the reviewer might misunderstand is that the point of the paper is not “higher energy needs a longer feature adaptation”, but “if we want the feature to adapt more, we need to use the designs that can provide more energy (like esHP, lsHP, partial backbone)”. Based on this, we can see what the non-trivial insights the theory provides. First, too much energy (i.e., HP 0 epochs) is usually harmful, as the direction term influences a lot. Under such a setting, the topological structure of all $z$s might change a lot, i.e., the model forgets the pretrained information a lot. Second, too little energy (i.e., HP for a long time) does not always help (this opposes the claims in Kumar et. al. 2021), as the features $z$ will change as examples in Figure 4. When the energy is appropriate, the $z$ vectors will first be stretched in a similar direction (an increase of $z_t^T z_0$ and a slight decrease of cos($z_t,z_0$) ), which makes the features more separable while the topological structure of the features doesn’t change too much. That is why we suggest to earlier stop the HP process (like when HP-train-acc reaches 80% or 90%, not 99%), as supported by experimental results in Figure4. We believe the idea of using $z_T^T z_0$, $\|z_T\|_2$, cos($z_T, z_0$), to draw a big picture of how $z$ adapts during fine-tuning is cool and non-trivial, as the previous works mainly focus on L2 distance, which only monotonically increases with the energy (remember the $z_T^T z_0$ and $\|z_T\|_2$ have quadratic form).

---

### Official Review · Reviewer_3J3Q · 2022-11-04

**Confidence:** 4
**Correctness:** 3
**Technical Novelty And Significance:** 3
**Empirical Novelty And Significance:** 3
**Recommendation:** 6

**Clarity, Quality, Novelty And Reproducibility:**

Clarity
=====

Fig 1, the authors mention that one of the ways the setup differs between their work and Kumar et al is that ‘information exchanges between the two parts of the network’. I don’t really understand this point. Doesn’t this information exchange always happen during finetuning? What is special here?

In Section 2.3, for the different degrees to which we want to change the feature extractor, in ‘strong’, what is meant by ‘the pretrained model overfits’? Not clear if this refers to overfitting to the upstream task or the downstream task, and how exactly this is measured.

Fig 1 caption: define X_{PT} and X_{DS} – I assume it’s ‘pretraining’ and ‘downstream’ respectively? But this should be stated.

Under Equation 1, in the equation z = Bx, I think this x should be X.

In Section 3.1, it’s not clear to me what the superscript 0 in z_t^{(0)} and x^{(0)} means. Is this the first dimension of the embedding or the first example in the batch or something else?

Figure 2: should explain the caption. I’m assuming this HP x means that the head probing was run for x epochs? Should clarify.

The title of Section 3.2 is “Initial value: two extreme cases”. I did not understand what this is referring to. Initial value of what? And what are the two extreme cases? Are they ‘strong’ and ‘tiny’? It’s confusing as the results shown and discussed in this section (e.g. in Figures 2 and 3) show FT results for various phases of HT tuning which if I understand correctly are the ‘intermediate’ cases?

Figure 4 caption says it’s clear that mild adaptation makes the features more separable. More separable than what? Looks like strong adaptation makes them even more separable. Also, this figure should clarify what is the task in this case. I assume we don’t always (i.e. for all tasks) expect greater separability with more adaptation. What does this depend on? Could we have the opposite phenomenon for other tasks?

I don’t understand Proposition 1. My understanding was that when the energy increases, the euclidean distance between original and final features also increases. But Proposition 1 states the opposite. Is this a mistake?

Figure 5: the names of the columns are hard to understand. I also don’t know what “1-gap” means. Why is the last column LP (linear probe?) instead of HP (head probe)? Not clear what exactly is the experimental setup here and what is the intended take-away.

The authors mention their analysis is also relevant for finding features that work well in a new, OOD domain. It would be useful to elaborate on this, as I feel if this is true it would add a lot of value to the paper. But I’m not currently able to see how it’s true.

Figure 6: the caption says HP but the x-axis of subplots says LP. which is it?

The tasks used in all tables should be described. All notations used in figures should be described. There were several other instances aside from the above where I wondered what exactly a symbol means.

Quality
======
In practice, it is not easy to judge to what degree we want to update the pretrained model for a new downstream task (‘strong’, ‘mild’ or ‘tiny’). This is due to often not having enough data or resources downstream to assess aspects like ‘the model overfits’, and in addition it is difficult to estimate task relatedness (e.g. how related or different is the downstream task to the upstream one) - this in and of itself is an open problem. So it’s not easy to tell whether the pretrained model is ‘reasonably good’ (for mild for instance) on the downstream task. Given this, it is not clear to me what is the recommendation given by the authors for how to modify the way in which we finetune models, since the recommendation seems to be conditional on somehow making this difficult choice.

While the paper proposes a set of practical tricks to alleviate this, like label smoothing, I’m not necessarily convinced that it always helps and never hurts (for all downstream tasks). For example, I see at least one entry in Table 1 where it can hurt performance, for the Flowers dataset, with Sup-C10. It feels like more analysis on the effect of label smoothing needs to be done before this recommendation can be safely made.

In Figure 3, it seems that the ‘end’ histogram always matches the e_y pretty well, regardless of the head used (random or after HT). This doesn’t really make a strong case for having to be careful about how we do HT / FT. Are there other cases where we can see bigger differences? Perhaps transferring to a downstream task that is “very different” from the upstream one?

While the analysis here is interesting, it’s not clear to me how exactly it relates to the ultimate goal: analyzing the effect of design choices of HP/FT on the actual downstream *performance*. As the authors also pointed out, analyzing the change on the features is an indirection and it’s less clear why it’s relevant / how to inform practical decision making.

Section 3.3 “backbone depth and head capacity” makes some intuitive observations which however aren’t too surprising, and also aren’t very actionable. Again, how can we assess whether low-level features (of the pretrained model) will be beneficial for a new downstream task while high-level features harmful?

Novelty
======
The proposed particular analysis is novel to the best of my knowledge.

Reproducibility
============
The procedure proposed is simple enough that I feel confident I could implement it. But I encourage the authors to provide all details about hyperparameters, task details etc, to aid in reproducibility, and make their code available.


**Strength And Weaknesses:**

Strengths
========
- To the best of my knowledge, this analysis of the influence of ‘energy’ in the head on the changes made to the features during FT is novel and it is interesting.

- The finding that adding label smoothing can make the HP-FT procedure more robust to the number of epochs for HP is an interesting one.

Weaknesses
==========
- I found that the paper has clarity issues (see below)

- Does label smoothing always help? Are there cases where this addition can degrade performance? (I see at least one such entry in Table 1, for Flowers dataset, Sup-C10). The authors don’t discuss the potential dangers of using it / weaknesses of this approach.

- I’m not sure what the practical take-aways are (see below)


**Summary Of The Paper:**

This paper studies the two-stage transfer learning approach of head tuning first (a generalization of linear probing where the head can be nonlinear), followed by finetuning both the backbone and continuing to finetune the head. Their work builds upon the setup of Kumar et al, but they claim to relax the assumption made in that work that the pretrained model is optimal for the downstream task.

They decompose the training update to the features during finetuning and show theoretically that when the adaptation ‘energy’ increases (meaning that the predictions on the downstream task differ from the correct labels), the final representations (obtained after finetuning) change more. So, intuitively, controlling that energy is one way of controlling how much adaptation is performed on the features. One way to increase the energy is by doing less head probing before the fine-tuning phase (this way the head will be less suited to the downstream task at the start of the finetuning phase, so more adaptation occurs). Since the number of head probing epochs is difficult to tune, they propose label smoothing as an additional knob that can affect energy increase and also explore the effect of the capacity of the head. They show in several empirical scenarios results consistent with their intuitions and theoretical results.


**Summary Of The Review:**

This paper analyzes the effect of different head designs on the adaptation that occurs in the learned features after the finetuning phase. Their analysis is novel to the best of my knowledge and some findings are interesting. However, I found that the writing and presentation can be improved (see clarity section above) and I’m also unsure about the usefulness of practical takeaways from the paper (see above detailed comments). For this reason, I’m doubtful if the size of the contribution meets the bar for acceptance.


===============================

After rebuttal: The new version of the paper is clearer and better structured. I particularly liked the User Guide section. I also like the discussion around difficulty in making these design choices in practice, the risks associated with some decisions (e.g. label smoothing, etc) and the inclusion of recommendations for making them based on increasingly-risky interventions. Based on this, I increased my score from a 5 to a 6.

---

> ### Author Response · Authors · 2022-11-13
> **Change the paper a lot, fix most clarity issues 1/3**
>
> Thank you for identifying the strengths of this work and sorry for the clarity issues in the previous version. We appreciate the thorough review and have made several enhancements based on your comments. If you have other concerns in mind, we would be happy to take them into consideration.
>
> Please also note the [**overall response**](https://openreview.net/forum?id=gVOXZproe-e&noteId=gSQsN_eyVRJ). Here is the piece-to-piece response.
>
> ### Weakness:
>
> > Does label smoothing always help? Are there cases where …
>
> Yeah, the label smoothing method does not always help. In this version, we re-arrange the proposed methods to one base method (sweeping $\tau$, which never hurts) and three advanced tricks (which all have their risks). The limitations of these tricks are elaborated on in section 5.3, including the label smoothing. We also recommend the reviewer refer to section 5.1, which highlights the takeaway message, to see when to consider these advanced tricks.
>
> > I’m not sure what the practical takeaways are (see below) …
>
> Thanks very much for pointing this out. We now highlight the practical takeaways in section 5.1.
>
> ### Clarity:
>
> > Fig 1, the authors mention that one of the ways the setup differs between their work …
>
> Sorry for the misleading words. We delete it in this version. The difference between this paper and Kumar’s are only two: 1. non-linear head; 2. non-optimal pretrained features. Btw, in Kumar’s assumption, in the ideal case, the information between the head and backbone won’t exchange, as the gradients of the backbone become zero.
>
> > In Section 2.3, for the different degrees to which we want to change the feature extractor …
>
> Thanks for pointing this out. We change the expression in this version. The ‘strong case’ refers to the scenarios we believe the pretrained features are far away from the optimal ones for the downstream tasks. Such a big gap might be caused by the pretrain overfitting, i.e., the backbone overfits the pretraining task and dataset seriously. Or when the downstream task is too different from the pretraining one. This is a rather vague description, as we cannot precisely define (or measure) how strong is strong. However, in section 4.2, we carefully discuss the difference between strong and mild cases. Please also refer to Q2 in the overall response.
>
> > Fig 1 caption: define $X_{PT}$ and $X_{DS}$ – I assume …
>
> Thanks for pointing this out, we fix this by adding the explanation in the caption.
>
> > Under Equation 1, in the equation z = Bx, I think this x should be X.
>
> Thanks for this comment. Here we use z=Bx to represent the function g(z). Both lowercase-bold letters z and x are for one sample. As we don’t denote the N-sample version of z (like X for x), so we decide to use the lowercase-bold letters here.
>
> > In Section 3.1, it’s not clear to me what the superscript 0 in $z_t^{(0)}$ and $x^{(0)}$ means …
>
> Sorry for the misleading superscript. We change the superscript to $z^{(j)}$ and explain what it means in the context. Actually, equation (2) describes how the representation of ANY sample $x^{(j)}$ changes (i.e., $z^{(j)}$) when the network parameters are updated by the gradients calculated by all the data samples (i.e., $x^{(n)}, n=1,…, N$).
>
> > Figure 2: should explain the caption. I’m assuming this HP x means that the head probing …
>
> Thanks for pointing out this. We add the explanation and fix all LP to HP.
>
> > The title of Section 3.2 is “Initial value: two extreme cases”.  …
>
> Sorry for the vague representation. We now change the name of these sections and rewrite them to make the paper clearer. For your concern: two extreme cases are ($\tau=0$ and $\tau=\infty$), which lead to strong and tiny respectively. As $\tau=\infty$ is impossible to reach, we use $\tau=1024$ (or 50000) to simulate it. It is also safe to consider all the $\tau>0$ cases as the intermediate cases, as $\tau=\infty$ (zero energy) is almost impossible in practice.
>
> > Figure 4 caption says it’s clear that mild adaptation makes the features more separable. …
>
> Thanks very much for raising this good question. We delete this imprecise description in this version (now it is Fig.3). We believe the current section 4.2 and Q2 in the overall response can answer this question well. In short, mild adaptation makes the features more separable than the tiny case, as the $\|z_T\|_2$ and $z_T^\top z_0$ will increase. This is beneficial for downstream generalization as there might be fewer overlapped features. For the more separable strong case, we claim that the feature’s manifold changes a lot (supported by PCA components and head-exchange experiment). We aim to use these experiments to explain the difference between strong, mild, and tiny cases, which we believe is non-trivial and cannot be concluded if we only focus on $\|z_T-z_0\|_2$. These experiments are in the Toy-MLP setting. We can add similar figures for the more practical task if the current one cannot persuade the readers.
>
> **TO BE CONTINUED**

---

> > ### Author Response · Authors · 2022-11-13
> > **Change the paper a lot, fix most clarity issues 2/3**
> >
> > > I don’t understand Proposition 1. My understanding …
> >
> > We feel very sorry about making such a mistake in the only proposition of this paper. We fix it in the current version (now it is proposition 2).
> >
> > > Figure 5: the names of the columns are hard to understand. …
> >
> > Sorry for the ambiguous captions, which make the paper hard to follow. Now they are all in the appendix. Anyway, the column titles are actually four distance-related measurements, i.e., $|z_T-z_0|, z_T^\top z_0, |z_T|_2$, and cos($z_T,z_0$). We will change them all in the camera-ready version, as generating these figures are kind of annoying. 1-gap is actually $1-\mathbb{E}\|e_y-p_0\|$. We will change them to $1-E_\text{aie}$, i.e., one minus the Average Initial Energy (AIE) in these figures. All LPs in the paper are changed to HP now. The experimental setup is stated in the names of these rows. The takeaway message for these figures is that our analysis and intuition about how z changes match the real settings (MLP, ResNet, GNN) well: we can see a similar trend of these quantities as those illustrated in the first panel in Figure3. The last column justifies that the HP time $\tau$ is a reasonable value to represent energy.
> >
> > > The authors mention their analysis is also relevant for finding features that work well in a new, OOD domain. …
> >
> > Thanks very much for pointing out this. We delete this sentence in the new version, as there is no space to discuss OOD in the main text. In the context of this paper, transfer learning already introduces a new distribution to study how features adapt — introducing another OOD distribution seems to help little. Actually, we are considering applying the analysis shown here in our future work. We can provide a simple example to see how this helps. Assume we want to compare the potential capability of two pretrained models on OOD tasks. If we can create 2-3 different supporting tasks (recall the meta-learning setting), and do HP$\tau$-FT on them, we can get 2-3 different optimal $\tau^*$. Based on our analysis, big $\tau^*$ means less energy is required when transferring to the supporting task. So we guess the good pretrained models should: 1.) have good supporting performance; 2.) on average, the $\tau^*$ should be large, which means the learned features need less adaptation for most of the downstream tasks. Not sure whether this high-level description could answer this question.
> >
> > > The tasks used in all tables should be described. All notations used in figures …
> >
> > Thanks very much for the suggestions. We update all the captions to make the tables and figures more readable in this version and provide detailed experimental settings in Appendix B.
> >
> > #### Quality:
> >
> > > In practice, it is not easy to judge to what degree we want to update the pretrained model …
> >
> > Yeah, we totally agree with the reviewer on this point. So we rearrange the paper and highlight the takeaway message in section 5 and Q1 in the overall response. Generally, selecting the hyperparameters and tricks for fine-tuning is quite a heuristic task. We usually start from guess, then try something, then tune something based on the results. On one hand, this paper provides more hyper-parameters and tricks to try. On another hand, the analysis here shed more light on why and when we should use these tricks. The theoretical analysis also ensures the generality of our suggestions. We also believe this work has the potential to inspire future works related to understanding the learning dynamics, features’ interoperability, etc.
> >
> > > While the paper proposes a set of practical tricks to alleviate this, like label smoothing, …
> >
> > Yeah, we thank the reviewer for pointing out this important issue. We add detailed discussions about when and why to use these advanced tricks (not only label smoothing) in section 5.3.
> >
> > > In Figure 3, it seems that the ‘end’ histogram always matches the e_y pretty well, …
> >
> > Thanks for the comments. We re-make the figures (now Fig.2) and re-write the explanation of this part in the current version. Specifically, the ‘start’ distribution (now $p_0$) is the prediction of the whole model at the beginning of FT while the ‘end’ distribution (now deleted) is that after FT. Considering one-hot supervision, the ‘end’ distribution would be quite close to the $e_y$ distribution. Because in most of the settings, the training accuracy can achieve almost 100% if we FT long enough. The concept of energy can be considered as the gap between the ‘start’ and the supervision’s distribution, i.e., ‘$e_y$’. A larger gap can lead to more adaptation of the pretrained features. Furthermore, we can also use the figures in this style to understand what happens in LS-HP.
> >
> > **TO BE CONTINUED**

---

> > > ### Author Response · Authors · 2022-11-13
> > > **Change the paper a lot, fix most clarity issues 3/3**
> > >
> > > > Section 3.3 “backbone depth and head capacity” makes some intuitive observations …
> > >
> > > Now they are in section 4.3. Yeah, the methods that only copying part of the pretrained model is more complex to analyze (but the concept of energy and adaptation still holds). We tried to formalize that but find it really difficult. It can be considered a heuristic method (sometimes quite helpful) in our toolbox. We consider it as an advanced trick and try it only if we have a strong belief. Please refer to section 5.3.
> > >
> > > > The procedure proposed is simple enough that I feel confident I could implement it. …
> > >
> > > Thanks for the comments. We add the details about the hyperparameters and tasks in the current version (Appendix B). The code will also be released latter.

---

> > > > ### Comment · Reviewer_3J3Q · 2022-11-18
> > > > **response to authors after rebuttal**
> > > >
> > > > Thank you for your response, clarifications and updates to the paper.
> > > > The new version of the paper is clearer and better structured. I particularly liked the User Guide section. I also like the discussion around difficulty in making these design choices in practice, the risks associated with some decisions (e.g. label smoothing, etc) and the inclusion of recommendations for making them based on increasingly-risky interventions.
> > > > Based on this, I increased my score to a 6.

---

### Author Response · Authors · 2022-11-13
**Overall response 1/2**

We thank the reviewers for their helpful feedback, which we think has led to significant improvements in the new revision of the paper. As many reviewers had similar concerns, we highlight common issues here.

## Q0: Highlight the change
**Change the flow of the paper (rewrite sections 2,3,4,5)**

Section2: we start from a counter-intuitive observation, i.e., stopping the training of the task head earlier than converge can improve the performance.

Section3: we provide the notations and raise the question that how much feature adaptation we expect during finetuning (this is a rather heuristic question, which will be elaborated on later in this response)

Section4: we analyze the relationship between $v_0$ and $z$’s adaptation step-by-step. By bounding the norm of $\|z_T-z_0\|$, we find the Average Initial Energy (defined in current proposition 1) is the pivot. By observing the big change in the ‘direction term’, we speculate that the difference between strong and mild adaptation is non-trivial. Hence we propose to analyze the other 3 similarity-related quantities using a simplified model in proposition 2. To verify our analysis, we propose the PCA experiment and head-exchange experiment (see Fig.3). Finally, we provide some high-level discussions about how features adapt when more complex and advanced tricks are applied (detailed discussions are in the experimental part).

Section5: we first provide a user guide, which highlights the takeaway message. Then we carefully discuss the main method (i.e., sweeping $\tau$) and three advanced tricks (MLP head, label smoothing, and partial backbone).

**The experiments and theory added (main context)**

a.) results on image segmentation in Fig.4, which further supports the generalizability of our analysis and methods.

b.) head-exchange experiment in Fig.3, which supports the claim that strong adaptation significantly changes the feature manifold while mild adaptation doesn’t.

c.) some supporting experiments to describe the benefits and side-effects when using the advanced tricks in section 5.3.

**The experiments and theory added (in the appendix)**

a.) the proof of Average Initial Energy (AIE), which is part of the upper bound of $\|z_T-z_0\|_2$.

b.) extend Appendix B for more detailed experimental settings (code will be released later).

**Some experiments in progressing (will add them if you think they are really helpful)**

a.) experiments supporting the claim that NTK changes slowly during finetuning. (As there are some papers claiming this, and the analysis in the current version doesn’t rely too much on this assumption, we left those experiments to a lower priority in the rebuttal phase.)

b.) some discussions and experiments about the OOD scenario. (We still think this cannot fit in the current flow of the main text, as the out-of-distribution validation set cannot influence the feature’s adaptation. We believe the transfer learning tasks already show how the features adapt to a different distribution — there is no need to add an extra distribution.)

c.) some discussions and experiments about the influence of downstream dataset size. (The current experiments contain downstream datasets with different sizes, but we cannot see any clear relationship between the influence of the downstream dataset size. Maybe their relationship is too complex, as claimed in the piecewise response to the reviewer kiZg. Hence we tend to surpass discussing this open problem in the current version.)

---

> ### Author Response · Authors · 2022-11-13
> **Overall response 2/2**
>
> ## Q1: Takeaway message and how to judge the PT/DS gap
>
> We highlight the takeaway message in section 5.1 (i.e., user guide). The goal of this paper is to provide a toolbox for better HP-FT design. Facing a practical problem, we can first speculate how much energy is required (yeah, defining the gap between PT/DS tasks is a rather hard, heuristic, and open problem and we cannot provide any precise knowledge about this before conducting experiments). Then we first suggest trying our basic method, i.e., sweeping $\tau$. If we have some stronger belief (e.g., the DS only cares about the edges while the PT has many texture features) or observe some clear signals during training (HP-train-acc converges to 100% too fast but we still need adaptation), we can try the three advanced tricks. Note that the advanced tricks all have their benefits and potential risks, which are elaborated on in section 5.3. The basic method almost never hurts (other than more electricity and CO2), as we need to run HP$\tau$-FT for different $\tau$ several times. We can use the validation set to select the hyperparameters and tricks provided here, just like sweeping the learning rate or deciding whether or not to use dropout.
>
> ## Q2: Non-trivial trend of z’s adaptation
>
> Many reviewers are curious about what is the “non-trivial trend” highlighted throughout the paper, so we explain it here. First, it is kind of trivial that larger energy leads to more adaptation, like the monotonicity between energy and $\|z_T-z_0\|_2$. However, as mentioned in section 4.2 and equation 2, such a trivial claim ignores the influence of the vector’s direction. If we consider the direction (i.e., the analysis on $z_T^\top z_0$, cosine and $\|z_T\|_2$), we can find some quadratic trend and hence imagine the adapted $z_T$ like the examples in Figure 3. Specifically, from tiny to mild, the feature’s manifold (or structure, or the relationship between features of different samples) changes little. We can consider the mild adaptation as searching for a better local optimum in the SAME BASIN as the initialized one. During this process, the originally overlapped features become more separable, which makes it easier for a linear head to distinguish different classes, and hence enhance the generalization ability. Comparing the strong and mild cases, even though the scatter plot of them are quite similar (the strong case make features even more separable), the feature’s manifold might already be changed. We have two supports: 1). the PCA component (i.e., the first two columns of the SVD matrix) are quite similar for the mild and tiny cases, but that of the strong case is quite different from them; 2). the head-exchange experiment: we can exchange the head between mild and tiny cases, but the features adapted in the strong case are not compatible with the heads from other cases (remember all the 49 settings can generalize well before head-exchange). In short, the strongly adapted features can solve the downstream task, but using quite different features, i.e., the strong adaptation is searching for a better local optimum in a DIFFERENT BASIN from the pretrained one: because in the noisy-direction phase, we might already jump out of the current basin as we have too much energy. Indeed, this non-trivial trend might only slightly influence the final performance. However, we believe our analysis and findings could be important for the works focusing on explainability or interpreting the features.

---

### Author Response · Authors · 2022-11-15
**Thanks for your comments. Is there any other additional points we should address?**

Dear AC and reviewers:

We appreciate the comments, suggestions, and feedback thus far. You have helped us to improve the communication of the technical and empirical contributions in the submission. We would be happy to follow up on your responses and look forward to addressing any outstanding points for discussion. Please let us know if there are additional points of clarification that you would like us to address. We appreciate your time and energy through the review process.

---

### Author Response · Authors · 2022-11-17
**Further update, experiments about the slow NTK changing assumption.**

Dear AC and reviewers:

We update our paper by fixing some typos and adding one small experiment in Appendix A. It empirically supports the assumption that the empirical NTK of the backbone will change slowly during finetuning. We observe that compared with training from scratch, copying a good pretrained model makes the training in the so-called NTK lazy-parameter regimen.

Also we would like to send a kindly reminder to the reviewers that the discussion only have one day left. We would be more than happy to continue the discussion and address any further concerns that you might have.

---

### Decision · Program_Chairs · 2023-01-20

**Decision:**

Accept: poster

**Justification For Why Not Higher Score:**

please see the meta review

**Justification For Why Not Lower Score:**

There has been significant improvement during the rebuttal period.

**Metareview: Summary, Strengths And Weaknesses:**

The paper analyzes how the choice of task head controls feature adaptation and hence influences the downstream performance.  By decomposing the feature's learning dynamics, they find the key aspect is the training accuracy and loss at the beginning of finetuning, which determines the "energy" available for the feature's adaptation.

Strength:
-  The analysis of the influence of ‘energy’ in the head on the changes made to the features during FT is novel and interesting.
-   Every consideration is presented both in a discussion and in a formal fashion
- Several comparisons are made with other similar analyses, outlining the differences in the proposed approach


Weakness:
- in practice NTK is slower and assuming being in the NTK regime weakens the paper
- applicability and clear takeaways for practitioners can be improved
- initial submission had a number of clarity issues that are mostly solved during the rebuttal period.
- non-trivial trend and gap estimation: it is not clear how the "non-trivial trend" helps to design the proposed of \tau sweeping strategy.
- It is not explained how optimal learning rate is selected for different tasks. It might indicate the necessity to search both learning rate


The paper was significantly changed during the rebuttal period and in response to the reviewers comments.

**Note From Pc:**

if the above contains the word "oral" or "spotlight" please see: "oral" presentation means -> notable-top-5% and "spotlight" means -> notable-top-25%. As stated in our emails, we are disassociating presentation type from AC recommendations

**Summary Of Ac-Reviewer Meeting:**

In the meetings reviewers mostly reiterated their reviews. The concern was raised that the paper is changed a lot during rebuttal and therefore, it is tough to make a detailed judgement on the quality of the final manuscript. None was super excited about the paper but at the same time they appreciated the effort authors put into the paper. In the end, still there was no strong sign of acceptance /reject.